

# Complete spectrum of quantum integrable lattice models associated to $Y(gl_n)$ by separation of variables

**Jean Michel Maillet[1⋆] and Giuliano Niccoli[1†]**

**1** Univ Lyon, Ens de Lyon, Univ Claude Bernard Lyon 1, CNRS,
Laboratoire de Physique, UMR 5672, F-69342 Lyon, France

⋆ maillet@ens-lyon.fr, † giuliano.niccoli@ens-lyon.fr

## Abstract

We apply our new approach of quantum Separation of Variables (SoV) to the complete characterization of the transfer matrix spectrum of quantum integrable lattice models associated to $gl_n$-invariant $R$-matrices in the fundamental representations. We consider lattices with $N$-sites and general quasi-periodic boundary conditions associated to an arbitrary twist matrix $K$ having simple spectrum (but not necessarily diagonalizable). In our approach the SoV basis is constructed in an universal manner starting from the direct use of the conserved charges of the models, e.g. from the commuting family of transfer matrices. Using the integrable structure of the models, incarnated in the hierarchy of transfer matrices fusion relations, we prove that our SoV basis indeed separates the spectrum of the corresponding transfer matrices. Moreover, the combined use of the fusion rules, of the known analytic properties of the transfer matrices and of the SoV basis allows us to obtain the complete characterization of the transfer matrix spectrum and to prove its simplicity. Any transfer matrix eigenvalue is completely characterized as a solution of a so-called quantum spectral curve equation that we obtain as a difference functional equation of order $n$. Namely, any eigenvalue satisfies this equation and any solution of this equation having prescribed properties that we give leads to an eigenvalue. We construct the associated eigenvector, unique up to normalization, of the transfer matrices by computing its decomposition on the SoV basis that is of a factorized form written in terms of the powers of the corresponding eigenvalues. Finally, if the twist matrix $K$ is diagonalizable with simple spectrum we prove that the transfer matrix is also diagonalizable with simple spectrum. In that case, we give a construction of the Baxter $Q$-operator and show that it satisfies a $T$-$Q$ equation of order $n$, the quantum spectral curve equation, involving the hierarchy of the fused transfer matrices.



# 1  Introduction

In this paper we continue our study of quantum integrable lattice models using the new approach to quantum separation of variables that we developed recently [1]. We use the framework of the quantum inverse scattering method [2–10]. In the present article we consider the class of quantum integrable lattice models associated to irreducible representations of the Yang-Baxter algebra obtained from tensor products of the fundamental representation corresponding to the $n^2 \times n^2$ rational $gl_n$-invariant $R$-matrices for any $n \geq 2$ [11–18]. The $gl_n$ symmetry of these $R$-matrices implies that the monodromy matrix multiplied by an arbitrary $n \times n$ twist matrix $K$ still satisfies the same Yang-Baxter algebra governed by the given $R$-matrix. Let us remark that for any choice of an invertible $K$-matrix, one gets a quantum integrable model associated in the homogeneous limit to the same bulk Hamiltonian as for $K = 1$ but having different $K$-dependent quasi-periodic boundary conditions.

Integrable quantum models in this class have been analyzed in the literature and exact results on the spectrum have been obtained e.g. by the use of generalizations of the standard algebraic Bethe ansatz like nested Bethe ansatz [11, 14] and analytic Bethe ansatz [15–18], with important recent progress towards their dynamics [19–28]. Here, we investigate them in the framework of the quantum Separation of Variable (SoV) approach pioneered by Sklyanin [29–34] along our new method of constructing an SoV basis presented in [1]. The SoV approach has in general the clear advantage to give a simple proof of the completeness of the spectrum description as it has been demonstrated in several important examples, mainly associated to the 6-vertex and 8-vertex representations of the Yang-Baxter and Reflexion algebras, see e.g. [29–70], which in Bethe ansatz framework is not an easy task in general. The SoV approach has been also shown to work for several integrable quantum models for which the

Algebraic Bethe ansatz cannot be directly used[1], due in particular to the absence of a so-called reference state, see e.g. [29]. It also allows to get some universal and straightforward simultaneous characterization of the transfer matrix eigenvalues and associated eigenvectors.

Let us recall that in the Sklyanin's SoV approach[2], the first step is to identify a one parameter commuting family of operators, the so-called $B$-operator, which must be diagonalizable and with simple spectrum. Then, let $Y_n$ be the operator zeroes of $B(\lambda)$. They form a set of commuting operators and their common eigenbasis can be labeled by their eigenvalues. Second, this $B$-operator family should have a "canonical conjugate" operator family, the so-called $A$-operator, also depending on a spectral parameter $\lambda$, which, thanks to the Yang-Baxter commutation relations, when carefully evaluated at $\lambda = Y_n$ acts as a shift operator over the spectrum of the $Y_n$. Third, the operator families $B(\lambda)$, $A(\lambda)$ and the transfer matrices of the model have to satisfy appropriate commutation relations implying that the operator families $A(\lambda)$ and the transfer matrices over the spectrum of the $Y_n$ satisfy a quantum spectral curve equation associated to the monodromy matrix $M(\lambda)$ satisfying a Yang-Baxter or Reflexion algebra. The fused transfer matrices appear there as operator coefficients and play the role of the quantum spectral invariants of the monodromy matrix $M(\lambda)$. As shown by Sklyanin, see e.g. [33], they are defined as quantum deformations of the corresponding classical spectral invariants. When these three steps are realized, the $Y_n$ are the so-called quantum separate variables for the transfer matrix spectral problem, the associated eigenbasis is the SoV basis and the separate relations are given by the quantum spectral curve equations that are finite difference equations over the spectrum of the separate variables.

Hence, this beautiful Sklyanin's picture for the construction of the SoV requires the identification of the operator families $B(\lambda)$ and $A(\lambda)$ and the proof that they satisfy all the outlined required properties. Sklyanin has proposed a way to identify these operator families[3] for a large class of models associated to the representation of the 6-vertex Yang-Baxter algebra and has implemented the procedure for some important models. As already mentioned, using his identification or simple generalization of it (see for example the idea of pseudo-diagonalizability of the $B$-operator family [59, 60]) it has been possible to widely implement the Sklyanin's SoV approach for integrable quantum model. Nevertheless, the Sklyanin's identification of the operator families $B(\lambda)$ and $A(\lambda)$ does not seem to be universal. In particular, for the higher rank cases, it appeared [1] that for the fundamental representation of the rational Yang-Baxter model associated to $gl_3$ it does not apply, as the proposed $A(\lambda)$ does not seem to act as a shift operator over the full $B$-spectrum.

Hence, until now, despite several important progress in their understanding [34, 37, 66, 77, 78], a systematic SoV description of higher rank quantum integrable models for a generic $K$-matrix has represented a longstanding open problem. Here, we solve it for the class of models associated to the fundamental representations of the Yangian $Y(gl_n)$ for general quasi-periodic boundary conditions associated to a matrix $K$ having simple spectrum (but not necessarily diagonalizable). This is done by implementing our new construction of the Separation of Variables (SoV) basis according to the general lines described in [1] where it was already applied to the cases $n = 2$ and $n = 3$.

The key point is that our SoV construction allows us to overcome the above mentioned problem of the identification of the operator families $B(\lambda)$ and $A(\lambda)$ and the proof of their required properties, e.g. the characterization of the $B$-spectrum and the proof of its diagonalizability and simplicity. In the case in which the Sklyanin's SoV approach works our SoV construction can be made to coincide with the Sklyanin's one by choosing appropriately our

---

[1]Note that, in the rank one case a modification of the original Algebraic Bethe Ansatz has been introduced to describe the XXZ and XXX quantum spin 1/2 chains with general integrable boundaries [72–76].

[2]At least for integrable quantum models associated to finite dimensional quantum spaces over finite lattices.

[3]On the basis of pure algebraic properties inferred by the Yang-Baxter commutation relations.

SoV-basis while our SoV construction applies to larger classes of models, as the higher rank cases that we are going to describe in this article.

In our approach [1] the SoV basis is constructed in an universal manner starting from the direct use of the conserved charges of the models, namely from the repeated action of the transfer matrices on a generic co-vector of the Hilbert space. The integrable structure of the model, incarnated in the transfer matrix fusion rules, are the basic tools used to prove the separation of the transfer matrix spectrum in our SoV basis. The complete characterization of the transfer matrix spectrum (eigenvalues and eigenvectors) is then obtained and its simplicity is proven. For any fixed eigenvalue the associated (unique up to normalization) eigenvector has coefficients in the SoV basis of factorized form written in terms of powers of the corresponding transfer matrix eigenvalues. The eigenvalues admit both a discrete and a functional equation characterization. Our SoV approach naturally leads to a characterization of the eigenvalues as the set of solutions to a system of $N$ (number of sites of the lattice) equations of order $n$ for $N$ unknowns. Then, in a second characterization, that we prove to be equivalent to the first one, they are obtained as the set of solutions of a functional equation which is an order $n$ finite difference functional equation. The coefficients of this quantum spectral curve equation are related to the quantum spectral invariant eigenvalues of the model, i.e. the eigenvalues of the fused transfer matrices. Finally, under the further condition that the twist matrix $K$ is diagonalizable with simple spectrum, we prove that the transfer matrices are also diagonalizable with simple spectrum. It allows us in this case to define a single higher rank analog of the Baxter $Q$-operator [79–102] satisfying with the transfer matrices the same order $n$ finite difference functional quantum spectral curve equation.

It is interesting here to make some further comments on the role played by the integrability in our SoV construction and on the subsequent characterization of the transfer matrix spectrum. We have recalled, in our previous paper [1], that the concept of independence of the charges, which is natural in classical integrability, is hard to define in the quantum case. Instead we have used there the requirement of $w$-simplicity (non-degeneracy) of the transfer matrix spectrum as an independence condition for generating the SoV basis. One has however to point out that from the $w$-simplicity of the spectrum of the commuting family of the transfer matrices it follows that we can always fix some value of the spectral parameter for which the corresponding transfer matrix, let us say $T(\lambda_0)$ is itself $w$-simple. Then, remarking that any operator commuting with a $w$-simple operator can be written as a polynomial of it of maximal degree $d-1$, with $d$ the dimension of the Hilbert space [103,104], one natural question that can emerge concerns the role of the other conserved charges and of the integrable structure. What we have shown in [1] is their role in the solution of the common spectral problem. Indeed, using only one $w$-simple operator $T(\lambda_0)$ we can in fact construct our basis according to formula (2.17) of our previous paper, so that the factorized characterization of the eigenvectors in terms of the eigenvalues of the Lemma 2.1 works. However this can be seen only as a pre-SoV characterization. Indeed, it leads to a characterization of the eigenvalues through a polynomial equation (given by the characteristic polynomial) of degree $n^N$ (the dimension of the Hilbert space) for $N$ sites. Instead, exploiting the full integrable structure of the model, in particular using the hierarchy of fused transfer matrices and their fusion relations, allows for the introduction of different type of spectrum characterization (discrete and functional) giving rise to equations of the quantum spectral curve of degree $n$.

Let us comment that the use of the fusion relations [12,14] in the framework of the quantum inverse scattering method to investigate the transfer matrix spectrum has already found several applications in the literature of quantum integrable models. A first systematic use of them has been introduced by Reshetikhin in his analytic Bethe ansatz method [15–18], see also [105,106]. There, these fusion relations are used to introduce an ansatz on the form of the transfer matrix eigenvalues which eventually leads to a nested system of Bethe equations

by the requirement of analyticity.

It is also interesting to remark that the connection with the fusion relations of the transfer matrices is evident already in the Sklyanin's SoV approach. This is intrinsically contained in the SoV as the condition of existence of transfer matrix eigenvectors [54–56] is just equivalent to the fusion relations of the transfer matrices computed in the spectrum of the separate variables and their shifted values. In [56] indeed it was explicitly stated, in the case of the 8-vertex model, that these fusion relations together with the known analyticity properties of the transfer matrix can be used to characterize the transfer matrix eigenvalues (as solutions to a system of quadratic equations of $N$ equations in $N$ unknowns). However, it was pointed out there that such a purely functional approach does not allow to identify the solutions to the system of equations which correspond to true eigenvalues. This is the case for the purely functional methods which do not allow for the construction of eigenvectors. Such a problem is indeed solved by our construction of the SoV basis in [1]. Indeed, our SoV basis with the combined use of transfer matrix fusion relations and their known analytic properties allows to identify the eigenvalues as the solutions to the system for which a unique (up to normalization) corresponding eigenvector can be constructed.

Finally, let us mention that while this work was already completed, an interesting paper [107] appeared, using the ideas of [1], also dealing with quantum integrable models associated to $gl_n$-invariant $R$-matrices. In this article, the conjecture discussed in [1] (and proven there for the $gl_2$ case) that our SoV basis can be constructed in such a way that it is an eigenbasis for the Sklyanin $B$-operator is argued for the $gl_n$ case. Let us comment however that the strategy we use in the present article (and also in [1]) to completely characterize the transfer matrix spectrum does not use the properties of the Sklyanin $B$-operator. Rather, as will be shown in the following, we use directly the properties of our SoV basis constructed from repeated actions of transfer matrices on a generic co-vector of the Hilbert space. From there, we will show that the structure constants of the commutative and associative Bethe algebra of conserved charges generated by the transfer matrices can be completely determined from the transfer matrix fusion relations and their asymptotic behavior properties (see eq.(3.20)). It leads in a direct way to the complete characterization of the spectrum of these transfer matrices (eigenvalues and corresponding eigenvectors). Notice that for Gaudin $gl_n$ models the interest of such a Bethe algebra was pointed out in [108]. Let us finally remark that it would be quite interesting to investigate the possible role of these structure constants in the approach of [109] to quantum integrable models.

This article is organized as follows. In section 2 we fix the basic definitions and the essential fusion and asymptotic behavior properties of the transfer matrices that we need for our purposes. In section 3 we completely characterize the spectrum of the transfer matrix for $gl_n$ models in the fundamental representations in terms of a discrete set of equations. In section 4 we prove that this characterization is equivalent to a functional quantum spectral curve equation. Further, in the case where the twist matrix is diagonalizable with simple spectrum we give a reconstruction of the Baxter $Q$-operator satisfying the corresponding $T$-$Q$ equation together with the fused transfer matrices. Finally in section 5 we give an Algebraic Bethe ansatz like rewriting of the transfer matrix complete spectrum. Important properties of the transfer matrices commutative algebra used in section 3 are proven in Appendix A. Similarly, technical proofs needed in section 4 are gathered in Appendix B.

## 2 Transfer matrices for quasi-periodic $Y(gl_n)$ fundamental model

Here and in the following we denote by $N$ the number of lattice sites of the model. Using the same notations as in [1], let us consider the Yangian $gl_n$ $R$-matrix

$$R_{a,b}(\lambda_a - \lambda_b) = (\lambda_a - \lambda_b)I_{a,b} + \eta\mathbb{P}_{a,b} \in \text{End}(V_a \otimes V_b), \quad \text{with } V_a = \mathbb{C}^n, V_b = \mathbb{C}^n, n \in \mathbb{N}^*, \quad (2.1)$$

where $\mathbb{P}_{a,b}$ is the permutation operator on the tensor product $V_a \otimes V_b$ and $\eta$ is an arbitrary complex number. It is solution of the Yang-Baxter equation written in $\text{End}(V_a \otimes V_b \otimes V_c)$:

$$R_{a,b}(\lambda_a - \lambda_b)R_{a,c}(\lambda_a - \lambda_c)R_{b,c}(\lambda_b - \lambda_c) = R_{b,c}(\lambda_b - \lambda_c)R_{a,c}(\lambda_a - \lambda_c)R_{a,b}(\lambda_a - \lambda_b), \quad (2.2)$$

and any matrix $K \in \text{End}(\mathbb{C}^n)$ satisfies:

$$R_{a,b}(\lambda_a, \lambda_b)K_a K_b = K_b K_a R_{a,b}(\lambda_a, \lambda_b) \in \text{End}(V_a \in V_b \otimes V_c), \quad (2.3)$$

i.e. it realizes the $gl_n$ invariance of the considered $R$-matrix. Then we can define the general (twisted) monodromy matrix,

$$M_a^{(K)}(\lambda) \equiv K_a R_{a,N}(\lambda_a - \xi_N) \cdots R_{a,1}(\lambda_a - \xi_1), \quad (2.4)$$

which satisfies the Yang-Baxter algebra,

$$R_{a,b}(\lambda_a - \lambda_b)M_a^{(K)}(\lambda_a)M_b^{(K)}(\lambda_b) = M_b^{(K)}(\lambda_b)M_a^{(K)}(\lambda_b)R_{a,b}(\lambda_a - \lambda_b), \quad (2.5)$$

in $\text{End}(V_a \otimes V_b \otimes \mathcal{H})$, with $\mathcal{H} \equiv \otimes_{l=1}^N V_l$ and its dimension $d = n^N$. Hence it defines a representation of the Yang-Baxter algebra associated to this $R$-matrix and the following one parameter family of commuting transfer matrices:

$$T^{(K)}(\lambda) \equiv \text{tr}_{V_a} M_a^{(K)}(\lambda). \quad (2.6)$$

In the above formulae, and in all this article, the complex parameters $\{\xi_1, ..., \xi_N\}$ are called *inhomogeneity parameters*, and we will assume in the following that they are in generic positions such that the above Yang-Baxter algebra representation is irreducible.

Let us define the following antisymmetric projectors:

$$P_{1,...,m}^- = \frac{\sum_{\pi \in S_m} (-1)^{\sigma_\pi} P_\pi}{m!}, \quad (2.7)$$

where:

$$P_\pi(v_1 \otimes \cdots \otimes v_m) = v_{\pi(1)} \otimes \cdots \otimes v_{\pi(m)}, \quad (2.8)$$

with $P_1^- = I$. Then the following proposition holds:

**Proposition 2.1** ([12, 13, 105]). *The fused transfer matrices (quantum spectral invariants):*

$$T_m^{(K)}(\lambda) \equiv \text{tr}_{1,...,m} \left[ P_{1,...,m}^- M_1^{(K)}(\lambda)M_2^{(K)}(\lambda - \eta) \cdots M_m^{(K)}(\lambda - (m-1)\eta) \right], \forall m \in \{1, ..., n\} \quad (2.9)$$

*generate n one parameter families of commuting operators:*

$$\left[ T_l^{(K)}(\lambda), T_m^{(K)}(\mu) \right] = 0, \qquad \forall l, m \in \{1, ..., n\}. \quad (2.10)$$

*The last quantum spectral invariant, the so-called quantum determinant:*

$$q{-}\text{det}M^{(K)}(\lambda) \equiv \text{tr}_{1,...,n} \left[ P_{1,...,n}^- M_1^{(K)}(\lambda)M_2^{(K)}(\lambda - \eta) \cdots M_n^{(K)}(\lambda - (n-1)\eta) \right],$$

*is moreover a central element of the algebra:*

$$[q{-}\text{det}M^{(K)}(\lambda), M_a^{(K)}(\mu)] = 0. \quad (2.11)$$

Moreover, the quantum spectral invariants satisfy the following properties:

**Proposition 2.2.** *The fused transfer matrices have the following polynomial form:*

*a) $T_m^{(K)}(\lambda)$ has degree $mN$ in $\lambda$ and central asymptotic behavior given by:*

$$T_m^{(K,\infty)} \equiv \lim_{\lambda \to \infty} \lambda^{-mN} T_m^{(K)}(\lambda) = \text{tr}_{1,\dots,m}\left[P_{1,\dots,m}^- K_1 K_2 \cdots K_m\right], \quad \forall m \in \{1,\dots,n-1\}, \quad (2.12)$$

*b) the quantum determinant reads:*

$$q-\det M^{(K)}(\lambda) = T_n^{(K)}(\lambda) = \det K \prod_{b=1}^{N}\left[(\lambda - \xi_b + \eta)\prod_{m=1}^{n-1}(\lambda - \xi_b - m\eta)\right]. \quad (2.13)$$

*The next fusion identities hold:*

$$T_1^{(K)}(\xi_a)T_m^{(K)}(\xi_a - \eta) = T_{m+1}^{(K)}(\xi_a), \quad \forall m \in \{1,\dots,n-1\}, \quad (2.14)$$

*together with the following central zeroes structure:*

$$T_m^{(K)}(\xi_a + r\eta) = 0, \quad \forall r \in \{1,\dots,m-1\}. \quad (2.15)$$

*Proof.* The general fusion identities [12, 13, 105] reduce to the above ones if computed in the inhomogeneities and their shifted values and moreover they imply the central zeroes structure stated in the proposition. $\qquad\square$

Let us introduce the functions

$$g_{a,\mathbf{h}}^{(m)}(\lambda) = \prod_{b \neq a, b=1}^{N} \frac{\lambda - \xi_b^{(h_b)}}{\xi_a^{(h_a)} - \xi_b^{(h_b)}} \prod_{b=1}^{N}\prod_{r=1}^{m-1} \frac{1}{\xi_a^{(h_a)} - \xi_b^{(-r)}}, \quad \xi_b^{(h)} = \xi_b - h\eta, \quad (2.16)$$

and

$$T_{m,\mathbf{h}}^{(K,\infty)}(\lambda) = T_m^{(K,\infty)} \prod_{b=1}^{N}\left(\lambda - \xi_b^{(h_n)}\right), \quad (2.17)$$

then the following corollary holds:

**Corollary 2.1.** *The transfer matrix $T_1^{(K)}(\lambda)$ allows to completely characterize all the higher transfer matrices $T_m^{(K)}(\lambda)$ by the fusion equations and central zeros structure in terms of the following interpolation formulae:*

$$T_{m+1}^{(K)}(\lambda) = \prod_{b=1}^{N}\prod_{r=1}^{m}(\lambda - \xi_b - r\eta)\left[T_{m+1,\mathbf{h}=\mathbf{0}}^{(K,\infty)}(\lambda) + \sum_{a=1}^{N} g_{a,\mathbf{h}=\mathbf{0}}^{(m+1)}(\lambda)T_m^{(K)}(\xi_a - \eta)T_1^{(K)}(\xi_a)\right], \quad (2.18)$$

*where we have denoted by $\mathbf{h} = \mathbf{0}$ the special set of values of $\mathbf{h}$ where for all $k$, $h_k = 0$.*

*Proof.* The known central zeroes and asymptotic behavior imply the above interpolation formula once we use the fusion equations to write $T_m^{(K)}(\xi_a)$. $\qquad\square$

# 3 Complete transfer matrix spectrum in our SoV approach

## 3.1 SoV covector basis for the quasi-periodic $Y(gl_n)$ fundamental model

The fundamental Proposition 2.4 proven in [1] for the construction of the SoV covector basis applies to the fundamental representation of the $gl_n$ rational Yang-Baxter algebra, i.e. the Yangian $gl_n$. Let us introduce the following notations

$$K = W_K K_J W_K^{-1}, \tag{3.1}$$

with $K_J$ the Jordan form of $K$ with the following block form:

$$K_J = \begin{pmatrix} K_J^{(1)} & 0 & \cdots & 0 \\ 0 & K_J^{(2)} & \ddots & 0 \\ 0 & \ddots & \ddots & 0 \\ 0 & 0 & \cdots & K_J^{(M)} \end{pmatrix}, \tag{3.2}$$

where any $K_J^{(a)}$ is a $d_a \times d_a$ Jordan block, let us say upper triangular, with eigenvalue $k_a$, where $\sum_{a=1}^{M} d_a = n$. Then Proposition 2.4 of [1] reads:

**Proposition 3.1** ([1]). *Let $K$ be a $n \times n$ w-simple matrix, i.e. a matrix with non-degenerate spectrum, then*

$$\langle h_1, ..., h_N | \equiv \langle S | \prod_{n=1}^{N} (T_1^{(K)}(\xi_n))^{h_n} \text{ for any } \{h_1, ..., h_N\} \in \{0, ..., n-1\}^{\otimes N}, \tag{3.3}$$

*form a covector basis of $\mathcal{H} = \bigotimes_{a=1}^{N} V_a$, with $V_a \simeq \mathbb{C}^n$, for almost any choice of $\langle S |$ and of the inhomogeneities satisfying the irreducibility condition. In particular, the state $\langle S |$ can take the next tensor product form:*

$$\langle S | = \bigotimes_{a=1}^{N} \langle S, a | \Gamma_W^{-1}, \quad \Gamma_W = \bigotimes_{a=1}^{N} W_{K,a}, \tag{3.4}$$

*where:*

$$\langle S, a | W_{K,a}^{-1} = (w_1^{(1)}, ..., w_{d_1}^{(1)}, w_1^{(2)}, ..., w_{d_2}^{(2)}, ..., w_1^{(M)}, ..., w_{d_M}^{(M)}) \in V_a, \tag{3.5}$$

*as soon as we take:*

$$\prod_{j=1}^{M} w_1^{(j)} \neq 0. \tag{3.6}$$

## 3.2 Transfer matrix spectrum characterization

Let us define the following $n-1$ polynomials in $\lambda$ and functions of a $n \times n$ matrix $K$ and of a point $\{x_1, ..., x_N\} \in \mathbb{C}^N$:

$$t_1^{(K,\{x\})}(\lambda) = \operatorname{tr} K \prod_{a=1}^{N} (\lambda - \xi_a) + \sum_{a=1}^{N} g_{a,\mathbf{h}=\mathbf{0}}^{(1)}(\lambda) x_a, \tag{3.7}$$

and from it:

$$t_{m+1}^{(K,\{x\})}(\lambda) = \prod_{b=1}^{N} \prod_{r=1}^{m} (\lambda - \xi_b - r\eta) \left[ T_{m+1,\mathbf{h}=\mathbf{0}}^{(K,\infty)}(\lambda) + \sum_{a=1}^{N} g_{a,\mathbf{h}=\mathbf{0}}^{(m+1)}(\lambda) x_a t_m^{(K,\{x\})}(\xi_a - \eta) \right], \tag{3.8}$$

for any $m \in \{1, ..., n-2\}$. Then, the following characterization of the transfer matrix spectrum holds:

**Theorem 3.1.** *Let us assume that the same conditions implying that* (3.3) *is a covector basis are satisfied, then we have the following characterization of the spectrum of* $T_1^{(K)}(\lambda)$:

$$\Sigma_{T^{(K)}} = \left\{ t_1(\lambda) : t_1(\lambda) = t_1^{(K,\{x\})}(\lambda), \quad \forall \{x_1, ..., x_N\} \in \Sigma_T \right\}, \tag{3.9}$$

$\Sigma_T$ *being the set of solutions to the following system of N equations of order n:*

$$x_a t_{n-1}^{(K,\{x\})}(\xi_a - \eta) = q-\det M^{(K)}(\xi_a), \tag{3.10}$$

*in N unknown* $\{x_1, ..., x_N\}$. *Furthermore, the spectrum of* $T_1^{(K)}(\lambda)$ *is non-degenerate and for any* $t_1(\lambda) \in \Sigma_{T^{(K)}}$ *the associated unique eigenvector* $|t\rangle$ *has the following wave-function in the left SoV basis, up-to an overall normalization:*

$$t_{1,\mathbf{h}} \equiv \langle h_1, ..., h_N | t \rangle = \prod_{a=1}^{N} t_1^{h_a}(\xi_a). \tag{3.11}$$

*Finally, if the n × n twist matrix K has simple spectrum and it is diagonalizable then* $T_1^{(K)}(\lambda)$ *is diagonalizable and with simple spectrum, for almost any choice of the inhomogeneity parameters.*

*Proof.* The system of $N$ equations of order $n$ in the $N$ unknown $\{x_1, ..., x_N\}$ coincides with the rewriting of the transfer matrix fusion equations:

$$t_1(\xi_a) t_{n-1}(\xi_a - \eta) = q-\det M^{(K)}(\xi_a), \quad \forall a \in \{1, ..., N\}, \tag{3.12}$$

for the eigenvalues of the transfer matrices $T_1^{(K)}(\lambda)$ and $T_{n-1}^{(K)}(\lambda)$. Moreover the recursion relations (3.8) just follow from the corresponding recursion relations for the fused transfer matrices in (2.18). So any eigenvalue of the transfer matrix has to satisfy this system of equations and the associated eigenvector $|t\rangle$ has the given characterization in the covector SoV basis.

The reverse statement has to be proven now. That is we have to prove that any polynomial $t_1(\lambda)$, of the above given form, which is solution of this system is an eigenvalue of the transfer matrix. We prove this showing that the vector $|t\rangle$ characterized by (3.11) is a transfer matrix eigenvector, i.e. that for any $\langle h_1, ..., h_N |$ it holds:

$$\langle h_1, ..., h_N | T_1^{(K)}(\lambda) | t \rangle = t_1(\lambda) \langle h_1, ..., h_N | t \rangle, \quad \forall \{h_1, ..., h_N\} \in \{0, ..., n-1\}^{\otimes N}. \tag{3.13}$$

This proof being quite lengthy, the main technical part of it is presented in the Appendix A. Let us just here give the main steps. The first remark is that the common spectrum of the transfer matrices evaluated in the $\xi_i$ being simple, any operator commuting with the transfer matrix can be obtained as a polynomial of maximal degree $n^N - 1$ (with $n^N$ the dimension of the Hilbert space) in the transfer matrix itself, see e.g., [103, 104]. It means that the vector space $\mathbf{B}_T$ (the so-called Bethe algebra) spanned by the operators commuting with the transfer matrices $T_m^{(K)}(\lambda)$ evaluated at an arbitrary spectral parameter $\lambda$ is of maximal dimension $n^N$. Now, let us denote by $T_{\mathbf{h}}^{(K)}$, with $\mathbf{h} = (h_1, ..., h_N)$ for any $\{h_1, ..., h_N\} \in \{0, ..., n-1\}^{\otimes N}$ the following products:

$$T_{\mathbf{h}}^{(K)} \equiv \prod_{n=1}^{N} (T_1^{(K)}(\xi_n))^{h_n}. \tag{3.14}$$

The fact that (3.3) defines a basis of our Hilbert space immediately implies that the system of operators given in (3.14) forms a free family of $n^N$ operators. Moreover any of these operators obviously commutes with the transfer matrix. Hence, the system of operators given in (3.14) forms a basis of the vector space $\mathbf{B}_T$ of operators commuting with the transfer matrix evaluated

at arbitrary values of the spectral parameter. As already noted above, to obtain the proof that the polynomial $t_1(\lambda)$ is an eigenvalue, we need to consider the action of $T_1^{(K)}(\xi_a)$, for $a = 1, \ldots, N$ on an arbitrary vector $\langle h_1, \ldots, h_N |$ of our SoV basis and to give its decomposition again on this basis. Then, the vector $|t\rangle$ being completely defined by its components on the SoV basis we can compute the necessary objects entering (3.13). Obviously, this amounts to be able to write the product $T_{\mathbf{h}}^{(K)} \cdot T_1^{(K)}(\xi_a)$ for any given $\mathbf{h}$ as a linear combination of $T_{\mathbf{h}'}^{(K)}$ with $\{h'_1, \ldots, h'_N\} \in \{0, \ldots, n-1\}^{\otimes N}$. However, since $T_{\mathbf{h}}^{(K)} \cdot T_1^{(K)}(\xi_a)$ is an operator commuting with any transfer matrix, it belongs to the vector space $\mathbf{B}_T$, and can be decomposed linearly on the basis given by the set (3.14). Namely, there exist sets of complex numbers $C_{\mathbf{h}\mathbf{h}'}^{(a)}$ depending on the two sets of parameters $\mathbf{h}$ and $\mathbf{h}'$ and on the index $(a)$ such that:

$$T_{\mathbf{h}}^{(K)} \cdot T_1^{(K)}(\xi_a) = \sum_{\mathbf{h}'} C_{\mathbf{h}\mathbf{h}'}^{(a)} T_{\mathbf{h}'}^{(K)}. \tag{3.15}$$

Then, using the interpolation formula for $T_1^{(K)}(\lambda)$, there exist sets of polynomials $C_{\mathbf{h}\mathbf{h}'}(\lambda)$ such that,

$$T_{\mathbf{h}}^{(K)} \cdot T_1^{(K)}(\lambda) = \sum_{\mathbf{h}'} C_{\mathbf{h}\mathbf{h}'}(\lambda) T_{\mathbf{h}'}^{(K)}. \tag{3.16}$$

Moreover, the results of Appendix A show that the set of complex numbers $C_{\mathbf{h}\mathbf{h}'}^{(a)}$ and hence of polynomials $C_{\mathbf{h}\mathbf{h}'}(\lambda)$ are completely determined by the hierarchy of fusion relations for the transfer matrices supplemented by their asymptotic behavior. Consequently, the above relation is also satisfied by the quantities $t_1(\xi_a)$, namely we have,

$$t_{1,\mathbf{h}} \cdot t_1(\xi_a) = \sum_{\mathbf{h}'} C_{\mathbf{h}\mathbf{h}'}^{(a)} t_{1,\mathbf{h}'}, \tag{3.17}$$

and,

$$t_{1,\mathbf{h}} \cdot t_1(\lambda) = \sum_{\mathbf{h}'} C_{\mathbf{h}\mathbf{h}'}(\lambda) t_{1,\mathbf{h}'}. \tag{3.18}$$

Hence we get for any choice of the co-vector $\langle h_1, \ldots, h_N |$,

$$\begin{aligned}
\langle h_1, \ldots, h_N | T_1^{(K)}(\lambda) | t \rangle &= \langle S | T_{\mathbf{h}}^{(K)} \cdot T_1^{(K)}(\lambda) | t \rangle \\
&= \langle S | \sum_{\mathbf{h}'} C_{\mathbf{h}\mathbf{h}'}(\lambda) T_{\mathbf{h}'}^{(K)} | t \rangle \\
&= \sum_{\mathbf{h}'} C_{\mathbf{h}\mathbf{h}'}(\lambda) t_{1,\mathbf{h}'} = t_{1,\mathbf{h}} \cdot t_1(\lambda) \\
&= t_1(\lambda) \langle h_1, \ldots, h_N | t \rangle.
\end{aligned} \tag{3.19}$$

This relation being true on the SoV basis, it proves that $|t\rangle$ is an eigenvector of the transfer matrix $T_1^{(K)}(\lambda)$ with eigenvalue $t_1(\lambda)$.

The final statement of the theorem about simplicity and diagonalizability of the transfer matrix has been already shown in Proposition 2.5 of [1]. □

Let us further remark that the above results have an interesting consequence. Indeed, using recursively the algebraic relation (3.15), one can obtain the following algebraic structure of the space $\mathbf{B}_T$:

$$T_{\mathbf{h}}^{(K)} \cdot T_{\mathbf{h}'}^{(K)} = \sum_{\mathbf{h}''} C_{\mathbf{h}\mathbf{h}'}^{\mathbf{h}''} T_{\mathbf{h}''}^{(K)}, \tag{3.20}$$

for a set of complex numbers coefficients $C_{\mathbf{h}\mathbf{h}'}^{\mathbf{h}''}$ that are completely determined (and computable) from the fusion relations satisfied by the transfer matrices. These coefficients are the structure constants of the associative and commutative algebra $\mathbf{B}_T$.

# 4 Transfer matrix spectrum by quantum spectral curve

## 4.1 The quantum spectral curve equation

The transfer matrix spectrum in our SoV basis is equivalent to the quantum spectral curve functional reformulation as stated in the next theorem.

**Theorem 4.1.** *Let the $n \times n$ matrix $K$ be nondegenerate with at least one nonzero eigenvalue. Then an entire function $t_1(\lambda)$ is a $T_1^{(K)}(\lambda)$ transfer matrix eigenvalue iff there exists a unique polynomial:*

$$\varphi_t(\lambda) = \prod_{a=1}^{\mathsf{M}}(\lambda - \lambda_a) \text{ with } \mathsf{M} \leq N \text{ and } \lambda_a \neq \xi_b \; \forall (a,b) \in \{1,...,\mathsf{M}\} \times \{1,...,N\}, \qquad (4.1)$$

*such that $t_1(\lambda)$,*

$$t_{m+1}(\lambda) = \prod_{b=1}^{N}\prod_{r=1}^{m}(\lambda - \xi_b - r\eta)\left[T_{m+1,\mathbf{h}=\mathbf{0}}^{(K,\infty)}(\lambda) + \sum_{a=1}^{N} g_{a,\mathbf{h}=\mathbf{0}}^{(m+1)}(\lambda)t_1(\xi_a)t_m(\xi_a - \eta)\right], \qquad (4.2)$$

*for any $m \in \{1,...,n-2\}$, and $\varphi_t(\lambda)$ are solutions of the following quantum spectral curve functional equation:*

$$\sum_{b=0}^{n}\alpha_b(\lambda)\varphi_t(\lambda - b\eta)t_{n-b}(\lambda - b\eta) = 0, \qquad (4.3)$$

*where we have defined*

$$t_0(\lambda) \equiv 1, \; t_n(\lambda) \equiv \det_q M^{(K)}(\lambda), \; \alpha_0(\lambda) \equiv -1, \qquad (4.4)$$

*and*

$$\alpha_1(\lambda) = \bar{\alpha}\, a(\lambda), \quad a(\lambda) = \prod_{a=1}^{N}(\lambda + \eta - \xi_a), \qquad (4.5)$$

$$\alpha_{1+j}(\lambda) = (-1)^j \prod_{h=0}^{j}\alpha_1(\lambda - h\eta), \quad \forall j \in \{1,...,n-1\}, \qquad (4.6)$$

*and $\bar{\alpha}$ is solution of the characteristic equation:*

$$\sum_{b=0}^{n}(-1)^{b+1}\bar{\alpha}^b T_{n-b}^{(K,\infty)} = 0, \qquad (4.7)$$

*i.e. $\bar{\alpha}$ is an eigenvalue of the matrix $K$. Moreover, up to a normalization the common transfer matrix eigenvector $|t\rangle$ admits the following separate representation:*

$$\langle h_1,...,h_N|t\rangle = \prod_{a=1}^{N}\alpha_1^{h_a}(\xi_a)\varphi_t^{h_a}(\xi_a - \eta)\varphi_t^{n-1-h_a}(\xi_a). \qquad (4.8)$$

*Proof.* Let the entire function $t_1(\lambda)$ satisfies with the polynomial $t_m(\lambda)$ and $\varphi_t(\lambda)$ the functional equation then it is a degree $N$ polynomial in $\lambda$ with leading coefficient $t_{1,N}$ solution of the equation:

$$\bar{\alpha}^n - \bar{\alpha}^{n-1}\, t_{1,N} + \sum_{b=0}^{n-2}(-1)^{b+1}\bar{\alpha}^b T_{n-b}^{(K,\infty)} = 0, \qquad (4.9)$$

which being $\bar{\alpha}$ an eigenvalue of $K$ implies:

$$t_{1,N} = \operatorname{tr} K. \tag{4.10}$$

Now for $\lambda = \xi_a$ it holds:

$$\alpha_{1+j}(\xi_a) = 0, \text{ for } 1 \le j \le n-1, \ \alpha_1(\xi_a) \neq 0, \ \det_q M^{(K)}(\xi_a) \neq 0, \tag{4.11}$$

and the functional equation reduces to:

$$\frac{\alpha_1(\xi_a)\varphi_t(\xi_a - \eta)}{\varphi_t(\xi_a)} = \frac{\det_q M^{(K)}(\xi_a)}{t_{n-1}(\xi_a - \eta)}. \tag{4.12}$$

While, for any fixed $s$ such that $1 \le s \le n-1$, for $\lambda = \xi_a + s\eta$ it holds:

$$\alpha_{r \ge s+2}(\xi_a + s\eta) = 0, \ t_{n-b}(\xi_a + (s-b)\eta) = 0, \text{ for any } 0 \le b \le s-1 \tag{4.13}$$
$$\alpha_{r \le s+1}(\xi_a + s\eta) \neq 0, \tag{4.14}$$

so that the functional equation reduces to:

$$\frac{\alpha_{s+1}(\xi_a + s\eta)\varphi_t(\xi_a - \eta)}{\alpha_s(\xi_a + s\eta)\varphi_t(\xi_a)} = \frac{t_{n-s}(\xi_a)}{t_{n-s-1}(\xi_a - \eta)}. \tag{4.15}$$

Now from the identities:

$$\frac{\alpha_{s+1}(\xi_a + s\eta)}{\alpha_s(\xi_a + s\eta)} = \alpha_1(\xi_a) \text{ for any } 1 \le s \le n-1, \tag{4.16}$$

the previous identities imply that the following equations are satisfied:

$$t_{m+1}(\xi_a) = t_m(\xi_a - \eta)t_1(\xi_a), \ \forall m \in \{1, ..., n-1\}, a \in \{1, ..., N\}, \tag{4.17}$$

so that by our previous theorem we have that $t_m(\lambda)$ are eigenvalues of the transfer matrices $T_m^{(K)}(\lambda)$, associated to the same eigenvector $|t\rangle$.

We derive now the reverse statement. That is, let $t_1(\lambda)$ be eigenvalue of the transfer matrix $T_1^{(K)}(\lambda)$ then we show that there exists a polynomial $\varphi_t(\lambda)$ satisfying with the $t_m(\lambda)$ the functional equation. The polynomial $\varphi_t(\lambda)$ is here characterized by imposing the next set of conditions:

$$\frac{\varphi_t(\xi_a - \eta)}{\varphi_t(\xi_a)} = \frac{t_1(\xi_a)}{\alpha_1(\xi_a)}. \tag{4.18}$$

The fact that this characterizes uniquely a polynomial of the form (4.1) is an essential point in the derivation of the functional equation starting from the SoV characterization of the spectrum and we have detailed it in Appendix B. Here, we prove that this characterization of $\varphi_t(\lambda)$ implies the validity of the functional equation. The l.h.s. of the functional equation is a polynomial in $\lambda$ of maximal degree $(n+1)N$ so to show that it is identically zero we have to prove it in $(n+1)N$ distinct points, being zero the leading coefficient by the choice of $\bar{\alpha}$ to be an eigenvalue of $K$. We use the following $(n+1)N$ points $\xi_a + k_a\eta$, for any $a \in \{1, ..., N\}$ and $k_a \in \{-1, 0, ..., n-1\}$. Indeed, for $\lambda = \xi_a - \eta$ it holds:

$$\alpha_r(\xi_a - \eta) = 0 \text{ for any } 1 \le r \le n, \text{ as well as } \det M^{(K)}(\xi_a - \eta) = 0, \tag{4.19}$$

from which the functional equation is satisfied for any $a \in \{1, ..., N\}$ and in the remaining $nN$ points the functional equation reduces to the $nN$ equations (4.12)-(4.15). Now, being the fusion equations satisfied by the transfer matrix eigenvalues, these equations are all equivalent to the discrete characterization (4.18) implying our statement.

Finally, we show that the SoV characterization of the transfer matrix eigenvectors is equivalent to that presented in this theorem, up to an overall normalization. Indeed, multiplying the eigenvector $|t\rangle$ by the non-zero product of the $\varphi_t^{n-1}(\xi_a)$ over all the $a \in \{1, \ldots, N\}$ it holds:

$$\prod_{a=1}^{N} \varphi_t^{n-1}(\xi_a) \prod_{a=1}^{N} t_1^{h_a}(\xi_a) \overset{(4.18)}{=} \prod_{a=1}^{N} \alpha_1^{h_a}(\xi_a) \varphi_t^{h_a}(\xi_a - \eta) \varphi_t^{n-1-h_a}(\xi_a). \tag{4.20}$$

$\square$

## 4.2 Reconstruction of the $Q$-operator and the Baxter $T$-$Q$ equation

The previous characterization of the transfer matrix spectrum indeed allows to reconstruct the $Q$-operator in terms of the elements of the monodromy matrix and more precisely in terms of the fundamental transfer matrix, as it is stated in the following:

**Corollary 4.1.** *Let us assume that $K$ is a $n \times n$ diagonalizable matrix with simple spectrum and let us denote with $\mathsf{k}_j$ the corresponding eigenvalues. Let us take*[4] *$\xi_{N+1} \neq \xi_{i \leq N}$ and let exist an $i \in \{1, \ldots, n\}$ such that $\mathsf{k}_i \neq 0$, then, for almost any values of $\{\xi_{i \leq N}\}$ and of $\{\mathsf{k}_{j \leq n}\}$, a $Q$-operator is given by the following polynomial family of commuting operators of maximal degree $N$:*

$$Q_i(\lambda) = \frac{\det_N[C_{i,\xi_{N+1}}^{(T_1^{(K)})} + \Delta_{\xi_{N+1}}(\lambda)]}{\det_N[C_{i,\xi_{N+1}}^{(T_1^{(K)})}]} \prod_{c=1}^{N} \frac{\lambda - \xi_c}{\xi_{N+1} - \xi_c}, \tag{4.21}$$

*where we have defined:*

$$[C_{i,\xi_{N+1}}^{(T_1^{(K)})}]_{rs} = -\delta_{rs} \frac{T_1^{(K)}(\xi_r)}{\mathsf{k}_i a(\xi_r)} + \prod_{\substack{c=1 \\ c \neq s}}^{N+1} \frac{\xi_r - \xi_c - \eta}{\xi_s - \xi_c} \qquad \forall r, s \in \{1, \ldots, N\}, \tag{4.22}$$

*and the central matrix of rank one:*

$$[\Delta_{\xi_{N+1}}(\lambda)]_{ab} = \frac{\lambda - \xi_{N+1}}{\xi_b - \lambda} \frac{\prod_{c=1}^{N}(\xi_c - \xi_a + \eta)}{\prod_{c=1, c \neq b}^{N+1}(\xi_c - \xi_b)} \qquad \forall a, b \in \{1, \ldots, N\}. \tag{4.23}$$

*Indeed, it satisfies with the transfer matrices the quantum spectral curve at operator level:*

$$\sum_{b=0}^{n} \alpha_b^{(i)}(\lambda) Q_i(\lambda - b\eta) T_{n-b}^{(K)}(\lambda - b\eta) = 0, \tag{4.24}$$

*where we have defined $T_0^{(K)}(\lambda) \equiv 1$ and the $\alpha_b^{(i)}(\lambda)$ are the polynomial coefficients defined in the previous theorem once we fix $\bar{\alpha} \equiv \mathsf{k}_i$, moreover $Q_i(\xi_a)$ are invertible operators for any $a \in \{1, \ldots, N\}$.*

*Proof.* This result is a direct consequence of the SoV characterization of the spectrum and of the proof of the previous theorem. As shown in Appendix B, for any $t_1(\lambda)$ eigenvalue of the transfer matrix $T_1^{(K)}(\lambda)$ we can associate the polynomial $\varphi_t(\lambda)$ of the form (4.1) solution of the quantum spectral curve equation with the transfer matrix eigenvalues. In that proof, we have shown that, up to an irrelevant overall nonzero normalization, $\varphi_t(\lambda)$ admits the following representation:

$$\varphi_t^{(i)}(\lambda) = \frac{\det_N[C_{i,\xi_{N+1}}^{(t_1)} + \Delta_{\xi_{N+1}}(\lambda)]}{\det_N[C_{i,\xi_{N+1}}^{(t_1)}]} \prod_{c=1}^{N} \frac{\lambda - \xi_c}{\xi_{N+1} - \xi_c}, \tag{4.25}$$

---

[4]Note that we can fix for example $\xi_{N+1} = \xi_h - \eta$ for any fixed $h \in \{1, \ldots, N\}$.

in terms of the above defined matrices where we have just replaced the transfer matrix $T_1^{(K)}(\xi_a)$ with the eigenvalues $t_1(\xi_a)$. In Proposition 2.5 of [1], we have also shown that for almost any value of the parameters $\{\xi_{i\leq N}\}$ and $\{k_{j\leq n}\}$ such that $K$ is diagonalizable and with simple spectrum then the transfer matrix $T_1^{(K)}(\lambda)$ is diagonalizable and with simple spectrum. This implies that we can uniquely define the polynomial operator family $Q_i(\lambda)$ by its action on the eigenbasis of the transfer matrix imposing:

$$Q_i(\lambda)|t\rangle = |t\rangle \varphi_t^{(i)}(\lambda), \qquad (4.26)$$

for any $t_1(\lambda)$ eigenvalue of the transfer matrix $T_1^{(K)}(\lambda)$ and $|t\rangle$ the uniquely (up to normalization) associated eigenstate. Then, by definition this operator family satisfies with the transfer matrices the quantum spectral curve equation, admits the announced representation in terms of the transfer matrix $T_1^{(K)}(\lambda)$ and the spectrum of its zeros never intersect the set of the $\{\xi_{i\leq N}\}$, which completes our proof. $\qquad\square$

*Remark* 4.1. The quantum spectral curve equation here derived follows in a quite constructive way in our SoV framework once one applies some intuitions inherited from the classical case. The fused transfer matrices are by definition the "shifted" quantum analogue of the classical spectral invariants. So, it is natural to look for them (and for their eigenvalues) to satisfy a "shifted" quantum analogous of the classical spectral curve equation. Then, we are left just with the determination of the shifts in the argument of these transfer matrices and the computation of the corresponding coefficients of the quantum spectral curve equations. As we have shown in the above theorem, these unknowns are indeed completely fixed by the known fusion equations and asymptotics of the transfer matrices.

# 5 Algebraic Bethe ansatz like rewriting of the spectrum

We can show that the previous SoV representation of the transfer matrix eigenvectors can be written in an Algebraic Bethe Ansatz form. Let us first observe that we can find one common eigenvector of the transfer matrices $T_m^{(K)}(\lambda)$ which correspond to the constant solution of the quantum spectral curve equation:

*Lemma* 5.1. *Let $K$ be a $n \times n$ $w$-simple matrix and let us denote with $K_J$ its Jordan form with $K = W_K K_J W_K^{-1}$, then:*

$$|t_0\rangle = \Gamma_W|0\rangle, \quad where \quad |0\rangle = \bigotimes_{a=1}^{N} \begin{pmatrix} 1 \\ 0 \\ \vdots \\ 0 \end{pmatrix}_a, \quad with \quad \Gamma_W = \bigotimes_{a=1}^{N} W_{K,a}, \qquad (5.1)$$

*is a common eigenvector of the transfer matrices $T_m^{(K)}(\lambda)$:*

$$T_1^{(K)}(\lambda)|t_0\rangle = |t_0\rangle t_{1,0}(\lambda), \quad with$$

$$t_{1,0}(\lambda) = k_1 \prod_{a=1}^{N}(\lambda - \xi_a + \eta) + (trK - k_1)\prod_{a=1}^{N}(\lambda - \xi_a), \qquad (5.2)$$

$$T_m^{(K)}(\lambda)|t_0\rangle = |t_0\rangle t_{m,0}(\lambda), \quad with$$

$$t_{m+1,0}(\lambda) = \prod_{b=1}^{N}\prod_{r=1}^{m}(\lambda - \xi_b - r\eta) \times \left[ T_{m+1,\mathbf{h}=\mathbf{0}}^{(K,\infty)}(\lambda) + \sum_{a=1}^{N} g_{a,\mathbf{h}=\mathbf{0}}^{(m+1)}(\lambda)t_{1,0}(\xi_a)t_{m,0}(\xi_a - \eta) \right], \qquad (5.3)$$

*where:*

$$K \begin{pmatrix} 1 \\ 0 \\ \vdots \\ 0 \end{pmatrix} = \mathsf{k}_1 \begin{pmatrix} 1 \\ 0 \\ \vdots \\ 0 \end{pmatrix} \quad \textit{with } \mathsf{k}_1 \neq 0 \tag{5.4}$$

*and where the $t_{m,0}(\lambda)$ satisfy the quantum spectral curve with constant $\varphi_t(\lambda)$:*

$$\sum_{b=0}^{n} \alpha_b(\lambda) t_{n-b,0}(\lambda - b\eta) = 0, \tag{5.5}$$

*for the choice $\bar{\alpha} = \mathsf{k}_1$.*

*Proof.* Note that the vector $|0\rangle$ is eigenvector of the transfer matrix $T_1^{(K_J)}(\lambda)$ with eigenvalue $t_{1,0}(\lambda)$. In fact, this is proven by showing that:

$$A_i^{(I)}(\lambda)|0\rangle = |0\rangle \prod_{a=1}^{N} (\lambda - \xi_a + \delta_{i,1}\eta), \quad C_i^{(I)}(\lambda)|0\rangle = 0, \quad i \in \{1, ..., n\}, \tag{5.6}$$

from which it easily follows that the vector $|0\rangle$ is eigenvector of all the others transfer matrices $T_m^{(K_J)}(\lambda)$ with eigenvalues $t_{m,0}(\lambda)$. Then, by the similarity relation:

$$T_1^{(K)}(\lambda) = \Gamma_W T_1^{(K_J)}(\lambda)\Gamma_W^{-1}, \tag{5.7}$$

we get our statement about the original transfer matrices. Note that these eigenvalues $t_{m,0}(\lambda)$ have to satisfy the quantum spectral curve as a consequence of the previous theorem. We have just to observe now that for the choice $\bar{\alpha} = \mathsf{k}_1$ it holds:

$$t_{1,0}(\xi_a) = \alpha_1(\xi_a) \quad \text{for any } a \in \{1, ..., N\}, \tag{5.8}$$

so that it follows that the associated $\varphi_t(\lambda)$ satisfies the equations:

$$\varphi_t(\xi_a) = \varphi_t(\xi_a - \eta) \quad \text{for any } a \in \{1, ..., N\} \tag{5.9}$$

and so $\varphi_t(\lambda)$ is constant. Indeed, defined:

$$\bar{\varphi}_t(\lambda) = \varphi_t(\lambda) - \varphi_t(\lambda - \eta); \tag{5.10}$$

this is a degree $N-1$ polynomial in $\lambda$ which is zero in $N$ different points so that it is identically zero. $\square$

Let us now define

$$\mathbb{B}^{(K)}(\lambda) = \prod_{a=1}^{N} \prod_{b=1}^{n-1} (\lambda - Y_a^{(b)}), \tag{5.11}$$

which is diagonal in the SoV basis and it is characterized by:

$$\langle h_1, ..., h_N | Y_a^{(b)} = (\xi_a - \eta\theta(b + h_a + 1 - n))\langle h_1, ..., h_N |, \tag{5.12}$$

from which it follows:

$$\langle h_1, ..., h_N | \mathbb{B}^{(K)}(\lambda) = b_{h_1, ..., h_N}(\lambda)\langle h_1, ..., h_N |, \tag{5.13}$$

where:

$$b_{h_1, ..., h_N}(\lambda) = \prod_{a=1}^{N} (\lambda - \xi_a)^{n-1-h_a} (\lambda - \xi_a + \eta)^{h_a}, \tag{5.14}$$

then the next corollary follows:

*Lemma* 5.2. *The following Algebraic Bethe Ansatz type formulation holds:*

$$|t\rangle = \prod_{a=1}^{\mathsf{M}} \mathbb{B}^{(K)}(\lambda_a)|t_0\rangle \quad \text{with } \mathsf{M} \le N \text{ and } \lambda_a \neq \xi_n \;\; \forall (a,n) \in \{1,...,\mathsf{M}\} \times \{1,...,N\}, \quad (5.15)$$

*for the eigenvector associated to the generic eigenvalue* $t_1(\lambda) \in \Sigma_{T_1}$. *Here the* $\lambda_a$ *are the roots of the polynomial* $\varphi_t(\lambda)$ *satisfying with the* $t_m(\lambda)$ *the quantum spectral curve functional equation.*

*Proof.* The following chain of identities holds:

$$\langle h_1,...,h_N| \prod_{a=1}^{\mathsf{M}} \mathbb{B}^{(K)}(\lambda_a)|t_0\rangle \;\; = \;\; \prod_{j=1}^{\mathsf{M}} b_{h_1,...,h_N}(\lambda_j)\, \langle h_1,...,h_N|t_0\rangle \tag{5.16}$$

$$= \;\; \prod_{j=1}^{\mathsf{M}} \prod_{a=1}^{N} (\lambda_j - \xi_a)^{n-1-h_a} (\lambda_j - \xi_a + \eta)^{h_a} \prod_{a=1}^{N} \alpha_1^{h_a}(\xi_a) \tag{5.17}$$

$$= \;\; \prod_{a=1}^{N} \alpha_1^{h_a}(\xi_a)\, \varphi_t(\xi_a)^{n-1-h_a}\, \varphi_t(\xi_a - \eta)^{h_a}, \tag{5.18}$$

where we have used that:

$$\langle h_1,...,h_N|t_0\rangle = \prod_{a=1}^{N} \alpha_1^{h_a}(\xi_a), \tag{5.19}$$

which coincides with the last SoV characterization of the same transfer matrix eigenvector. $\qquad\square$

# 6  Conclusion

In this article we have shown how to solve higher rank $gl_n$ quantum integrable lattice models using our new SoV approach [1]. The key ingredient is provided by the commutative Bethe algebra of conserved charges with structure constants following from the integrable Yang-Baxter algebra and encoded in the fusion relations for the transfer matrices. As in our previous analysis by SoV approach of quantum integrable models, the general idea is to completely characterize the spectrum, and eventually the dynamics, first in the inhomogeneous case. The corresponding results for the homogeneous models have then to emerge by taking the homogeneous limit, namely, the limit where all inhomogeneity parameters are set equal to some particular value, such that one recover the Hamiltonians of the homogeneous models. It should be noted that the characterization of the transfer matrix spectrum by the quantum spectral curve equation has a smooth homogeneous limit. So starting from the complete characterization of the transfer matrix spectrum in the inhomogeneous model one can get their homogeneous limit. However, one still has to prove that in this homogeneous limit the characterization remains complete, i.e., that the full spectrum is described by the corresponding quantum spectral curve equation. One way to prove it can be by direct construction of the SoV basis in the homogeneous models. This requires the proof of the existence of a set of commuting conserved charges with common non-degenerate spectrum. Then, a construction of an SoV basis is made possible in our approach. In particular, we can extract these conserved charges from the full set of fused transfer matrices if one can prove that they still have a common simple spectrum in the homogeneous limit. Once this key feature is achieved we have to find yet the separate relations characterizing the spectrum. They should once again be derived from the fusion relations, the quantum determinant centrality condition and the characterization of the higher fused transfer matrices in terms of the fundamental transfer matrix. The analysis of these interesting points is currently under study and their resolution could represent an

important steps towards the proof of the completeness of the spectrum characterization for the homogeneous integrable quantum models.

About the homogeneous limit of the transfer matrix eigenvectors, it is worth recalling that up to a scalar factor our $\mathbb{B}$-operator (5.11) should coincide [1, 107] with the Sklyanin's $B$-operator for an appropriate choice of our SoV basis. The fact that Sklyanin's $B$-operator admits a generic writing in terms of the elements of the monodromy matrix which is independent w.r.t. the inhomogeneities makes this observation important. Indeed, we have shown by SoV that any transfer matrix eigenvector can be rewritten in an ABA form in terms of this $\mathbb{B}$-operator computed in the zeroes of the $\varphi$-functions acting over the reference vector $|t_0\rangle$ (5.1). This type of ABA rewriting of the transfer matrix eigenvectors is well adapted to take smoothly the homogeneous limit. Indeed, the form of the Sklyanin's $B$-operator and of the reference vector are unchanged under this limit while the $\varphi$-function is now solution of the quantum spectral curve equation in the homogeneous limit, which is well defined too. However, while in the inhomogeneous model these vectors are known to be nonzero and independent transfer matrix eigenvectors, as a consequence of the existence of the SoV basis, one has still to prove that the same is true in this homogeneous limit. Once this is shown they are transfer matrix eigenvectors in the homogeneous limit associated to the corresponding transfer matrix eigenvalues satisfying with the $\varphi$-function the quantum spectral curve equation. The construction of the SoV basis directly in the homogeneous limit can be one possible way to prove this statement. An other way can be the analysis and the computation of scalar product and norms of separate vectors (that include all transfer matrix eigenvectors) first in the inhomogeneous models and then in the homogeneous limit. In the SoV framework, for the rank one related models, this type of analysis has been already developed with our collaborators N. Kitanine and V. Terras [65, 67, 71]. The extension of these works to the higher rank case is the fundamental first step toward the determination of the dynamics of these integrable quantum models in the SoV framework.

## Acknowledgements

J. M. M. and G. N. are supported by CNRS and ENS de Lyon.

## A    Appendix A

This appendix is dedicated to complete the proof of the Theorem 3.1. In the first two subsections, we first introduce some partial results and some tools to compute the action of transfer matrices on the elements of the covector SoV basis then we use them in the third subsection to complete the proof of the Theorem 3.1.

### A.1    Transfer matrix action on inner covectors of the SoV basis

Let us start introducing some notations for the SoV covector basis (3.3) for our current aims, we denote:

$$\langle h_1, ..., h_N| = \langle N_1^{\{h\}}, ..., N_n^{\{h\}}|, \tag{A.1}$$

where

$$N_j^{\{h\}} = \sum_{l=1}^{N} \delta_{h_l, j-1} \quad \forall j \in \{1, ..., n\} \tag{A.2}$$

and in the following we suppress the index $\{h\}$ for brevity. Moreover, we introduce the further notations:

$$N_m^- = \sum_{a=0}^{n-m-1}(n-m-a)N_{1+a}, \quad N_m^+ = \sum_{a=1}^{m} aN_{n-m+a} \tag{A.3}$$

for any $1 \le m \le n$, and we use the shorted notation

$$\langle h_1, ..., h_N | = \langle (N_m^-, N_m^+)|, \tag{A.4}$$

for the generic element of the SoV covector basis when all the information that we need to know are contained in this two numbers. Then, the following lemma holds:

*Lemma* A.1. *Under the same assumption of the Theorem 3.1, the following identities hold:*

$$\langle (N_m^-, N_m^+)|T_m^{(K)}(\lambda)|t\rangle = t_m(\lambda)\langle(N_m^-, N_m^+)|t\rangle, \forall m \in \{1, ..., n-1\}, \tag{A.5}$$

*if either $N_m^- = 0$ or $N_m^+ = 0$.*

*Proof.* Let us start observing that for $h_a \le n-2$ and $h_b \in \{0, ..., n-1\}$, for any $b \in \{1, ..., N\}\backslash a$, it holds:

$$\begin{aligned}\langle h_1, ..., h_N |T_1^{(K)}(\xi_a)|t\rangle &= \langle h_1, ..., h_a+1, ..., h_N|t\rangle \\ &= t_1(\xi_a)\langle h_1, ..., h_a, ..., h_N|t\rangle,\end{aligned} \tag{A.6}$$

as a direct consequence of the definition of the covector SoV basis and of the state $|t\rangle$. Then, the above identity implies the following ones:

$$\langle N_1, ..., N_{n-1}, N_n = 0|T_1^{(K)}(\xi_a)|t\rangle = t_1(\xi_a)\langle N_1, ..., N_{n-1}, N_n = 0|t\rangle, \ \forall a \in \{1, ..., N\}, \tag{A.7}$$

and so, being $t_1(\lambda)$ a polynomial of degree $N$ with known asymptotics which coincides by definition with the central one of $T_1^{(K)}(\lambda)$, it holds also:

$$\langle N_1, ..., N_{n-1}, N_n = 0|T_1^{(K)}(\lambda)|t\rangle = t_1(\lambda)\langle N_1, ..., N_{n-1}, N_n = 0|t\rangle \ \ \forall \lambda \in \mathbb{C}. \tag{A.8}$$

Let us remark now that the interpolation formulae (2.18), for the higher transfer matrices $T_m^{(K)}(\lambda)$, and the formulae (3.8), for the higher functions $t_m(\lambda)$, can be rewritten as it follows:

$$T_m^{(K)}(\lambda) = \prod_{b=1}^{N}\prod_{r=1}^{m-1}(\lambda - \xi_b - r\eta)\left[T_{m,\mathbf{h=0}}^{(K,\infty)}(\lambda) + \sum_{1 \le a_1 \le \cdots \le a_m \le N} r_{a_1,...,a_m}(\lambda)\prod_{i=1}^{m}T_1^{(K)}(\xi_{a_i})\right], \tag{A.9}$$

$$t_m(\lambda) = \prod_{b=1}^{N}\prod_{r=1}^{m-1}(\lambda - \xi_b - r\eta)\left[T_{m,\mathbf{h=0}}^{(K,\infty)}(\lambda) + \sum_{1 \le a_1 \le \cdots \le a_m \le N} r_{a_1,...,a_m}(\lambda)\prod_{i=1}^{m}t_1(\xi_{a_i})\right], \tag{A.10}$$

where $r_{a_1,...,a_m}(\lambda)$ are some computable by induction polynomials of degree one in $\lambda$, whose explicit values are not required for the following. From these formulae it follows that:

$$\begin{aligned}\langle N_1, ..., N_{n-m}, N_{n-m+1} = 0, ..., N_n = 0|T_m^{(K)}(\lambda)|t\rangle = \\ = t_m(\lambda)\langle N_1, ..., N_{n-m}, N_{n-m+1} = 0, ..., N_n = 0|t\rangle,\end{aligned} \tag{A.11}$$

for any $\lambda \in \mathbb{C}$ and $m \in \{1, ..., n-1\}$.

Similarly, let $1 \le h_a$ and $h_b \in \{0, ..., n-1\}$ for any $b \in \{1, ..., N\} \backslash a$, then it holds:

$$\langle h_1, ..., h_N | T_{n-1}^{(K)}(\xi_a - \eta) | t \rangle = q\text{-det}M^{(K)}(\xi_a) \langle h_1, ..., h_a - 1, ..., h_N | t \rangle$$
$$= \frac{q - \text{det}M^{(K)}(\xi_a)}{t_1(\xi_a)} \langle h_1, ..., h_a, ..., h_N | t \rangle$$
$$= t_{n-1}(\xi_a - \eta) \langle h_1, ..., h_a, ..., h_N | t \rangle. \tag{A.12}$$

Then, the above identity implies that it holds:

$$\langle N_1 = 0, N_2, ..., N_n | T_{n-1}^{(K)}(\xi_a - \eta) | t \rangle = t_{n-1}(\xi_a - \eta) \langle N_1 = 0, N_2, ..., N_n | t \rangle \ \ \forall a \in \{1, ..., N\} \tag{A.13}$$

and so also:

$$\langle N_1 = 0, N_2, ..., N_n | T_{n-1}^{(K)}(\lambda) | t \rangle = t_{n-1}(\lambda) \langle N_1 = 0, N_2, ..., N_n | t \rangle \ \ \forall \lambda \in \mathbb{C}. \tag{A.14}$$

Finally, it is easy to prove by induction that it holds:

$$\langle N_1 = 0, ..., N_{n-m} = 0, N_{n-m+1}, ..., N_n | T_m^{(K)}(\lambda) | t \rangle =$$
$$t_m(\lambda) \langle N_1 = 0, ..., N_{n-m} = 0, N_{n-m+1}, ..., N_n | t \rangle. \tag{A.15}$$

Indeed, let us assume that it holds for $m \le n-1$ and let us prove it for $m-1$, for any $a \in \{1, ..., N\}$ we have that:

$$\langle N_1 = 0, ..., N_{n-m} = 0, N_{n+1-m} = 0, N_{n-m+2}, ..., N_n | T_{m-1}^{(K)}(\xi_a - \eta) | t \rangle$$
$$= \langle N_1 = 0, ..., N_{n-m} = 0, N_{n+1-m} = \delta_{n-m+1}^{h_a}, N_{n-m+2} - \delta_{n-m+1}^{h_a} + \delta_{n-m+2}^{h_a}, ..., N_n - \delta_{n-1}^{h_a} | T_m^{(K)}(\xi_a) | t \rangle$$
$$= t_m(\xi_a) \langle N_1 = 0, ..., N_{n-m} = 0, N_{n+1-m} = \delta_{n-m+1}^{h_a}, N_{n-m+2} - \delta_{n-m+1}^{h_a} + \delta_{n-m+2}^{h_a}, ..., N_n - \delta_{n-1}^{h_a} | t \rangle$$
$$= t_{m-1}(\xi_a - \eta) \langle N_1 = 0, ..., N_{n-m} = 0, N_{n+1-m} = 0, N_{n-m+2}, ..., N_n | t \rangle, \tag{A.16}$$

from which our statement holds. $\qquad\square$

## A.2 Transfer matrix action on not inner covectors of the SoV basis

So to prove the Theorem 3.1, we are left with the proof of the above identities in the case both $N_m^-$ and $N_m^+$ are nonzero. In order to prove this we have to show that starting from matrix elements of the type

$$\langle \bar{N}_1, ..., \bar{N}_n | T_m^{(K)}(\lambda) | t \rangle, \tag{A.17}$$

and

$$t_m(\lambda) \langle \bar{N}_1, ..., \bar{N}_n | t \rangle, \tag{A.18}$$

with $N_m^-(\{\bar{N}_1, ..., \bar{N}_n\}) = \bar{N}_m^-$ and $N_m^+(\{\bar{N}_1, ..., \bar{N}_n\}) = \bar{N}_m^+$ both nonzero, we can develop simultaneously these matrix elements leading to linear combinations of the type:

$$\langle \bar{N}_1, ..., \bar{N}_n | T_m^{(K)}(\lambda) | t \rangle = \sum_{r=1}^{n-1} \sum_{a=1}^{N} \sum_{h=0}^{1} \sum_{\substack{\{N_1, ..., N_n\} \in \\ S_{r,a,h}^{(m,\{\bar{N}\})}}} C_{\{N_1,...,N_n\}}^{(m,\{\bar{N}\},r,a,h)}(\lambda) \langle N_1, ..., N_n | T_r^{(K)}(\xi_a^{(h)}) | t \rangle, \tag{A.19}$$

and

$$t_m(\lambda) \langle \bar{N}_1, ..., \bar{N}_n | t \rangle = \sum_{r=1}^{n-1} \sum_{a=1}^{N} \sum_{h=0}^{1} \sum_{\substack{\{N_1, ..., N_n\} \in \\ S_{r,a,h}^{(m,\{\bar{N}\})}}} C_{\{N_1,...,N_n\}}^{(m,\{\bar{N}\},r,a,h)}(\lambda) t_r(\xi_a^{(h)}) \langle N_1, ..., N_n | t \rangle, \tag{A.20}$$

but now with covectors satisfying the conditions:

$$N_m^-(\{N_1, ..., N_n\})N_m^+(\{N_1, ..., N_n\}) = 0 \quad \forall \{N_1, ..., N_n\} \in S_{r,a,h}^{(m,\{\bar{N}\})}, \tag{A.21}$$

where for any fixed $m, \{\bar{N}\}, r, a, h$ the $S_{r,a,h}^{(m,\{\bar{N}\})}$ is some given set of compatible $n$-tuples with the definitions of the integers $N_i$. Indeed, if proven such developments imply the theorem by the previous lemma. The fact that the coefficients $C_{\{N_1,...,N_n\}}^{(m,\{\bar{N}\},r,a,h)}(\lambda)$ of the above developments are the same, will follow from the fact that we make exactly the same type of operations on the two type of matrix elements, as described in the next.

### A.2.1 Interpolation expansions and fusion properties

First, we introduce the following rules to use the interpolation formulae

$$T_m^{(K)}(\lambda) = \prod_{b=1}^{N}\prod_{r=1}^{m-1}(\lambda - \xi_b - r\eta)\left[T_{m,\Delta^{(m)}}^{(K,\infty)}(\lambda) + \sum_{a=1}^{N} g_{a,\Delta^{(m)}}^{(m)}(\lambda)T_m^{(K)}(\xi_a^{(\Delta_a^{(m)})})\right], \tag{A.22}$$

and

$$t_m(\lambda) = \prod_{b=1}^{N}\prod_{r=1}^{m-1}(\lambda - \xi_b - r\eta)\left[T_{m,\Delta^{(m)}}^{(K,\infty)}(\lambda) + \sum_{a=1}^{N} g_{a,\Delta^{(m)}}^{(m)}(\lambda)t_m(\xi_a^{(\Delta_a^{(m)})})\right], \tag{A.23}$$

where the $N$-tupla $\Delta^{(m)} = \{\Delta_1^{(m)}, ...., \Delta_N^{(m)}\}$ is chosen according to the covector $\langle h_1, ..., h_N|$ as it follows:

$$\Delta_a^{(m)} = \begin{cases} 0 \text{ if } 0 \le h_a \le n-1-m \\ 1 \text{ if } n-m \le h_a \le n-1 \end{cases}. \tag{A.24}$$

Second, we use the fusion equations to rewrite

$$\langle \bar{N}_1, ..., \bar{N}_n|T_m^{(K)}(\xi_a^{(\Delta_a^{(m)})})|t\rangle, \tag{A.25}$$

and

$$t_m(\xi_a^{(\Delta_a^{(m)})})\langle \bar{N}_1, ..., \bar{N}_n|t\rangle, \tag{A.26}$$

at it follows. If $0 \le h_a \le n-1-m$, then they admit the following rewriting:

$$\langle N_1, ..., N_n|T_{m-1}^{(K)}(\xi_a^{(1)})|t\rangle, \tag{A.27}$$

and

$$t_{m-1}(\xi_a^{(1)})\langle N_1, ..., N_n|t\rangle, \tag{A.28}$$

where:

$$N_1 = \bar{N}_1 - \delta_0^{h_a}, \quad N_j = \bar{N}_j + \delta_{j-2}^{h_a} - \delta_{j-1}^{h_a} \text{ for } 2 \le j \le n-m \tag{A.29}$$

$$N_{n+1-m} = \bar{N}_{n+1-m} + \delta_{n-m-1}^{h_a}, \quad N_f = \bar{N}_f \text{ for } n+2-m \le f \le n. \tag{A.30}$$

Now denoted with $N_{m-1}^-$ and $N_{m-1}^+$ the integers associated to the above covectors, it holds:

$$N_{m-1}^- = \bar{N}_{m-1}^- - 1, \quad N_{m-1}^+ = \bar{N}_{m-1}^+, \tag{A.31}$$

where $\bar{N}_{m-1}^-$ and $\bar{N}_{m-1}^+$ are the integers associated to the covector $\langle \bar{N}_1, ..., \bar{N}_n|$. While, if $n-m \le h_a \le n-1$, then we have the following rewriting:

$$\langle N_1, ..., N_n|T_{m+1}^{(K)}(\xi_a)|t\rangle, \tag{A.32}$$

and

$$t_{m+1}(\xi_a)\langle N_1,...,N_n|t\rangle, \tag{A.33}$$

where:

$$N_j = \bar{N}_j \quad \text{for } 1 \leq j \leq n-1-m, \quad N_{n-m} = \bar{N}_{n-m} + \delta_{n-m}^{h_a} \tag{A.34}$$

$$N_f = \bar{N}_f - \delta_{f-1}^{h_a} + \delta_f^{h_a} \text{ for } n+1-m \leq f \leq n-1, \quad N_n = \bar{N}_n - \delta_{n-1}^{h_a}, \tag{A.35}$$

and now it holds:

$$N_{m+1}^- = \bar{N}_{m+1}^-, \quad N_{m+1}^+ = \bar{N}_{m+1}^+ - 1. \tag{A.36}$$

The following lemma will be used in the proof of the Theorem 3.1:

*Lemma A.2. Let us define $N_m^T \equiv N_m^- + N_m^+$, then it holds*

$$N \leq N_m^T \equiv N_m^- + N_m^+ \leq N \times \max\{m, n-m\} \tag{A.37}$$

*with the condition*

$$N_m^T = N \quad \text{if and only if} \quad N_{n-m} + N_{n+1-m} = N, \tag{A.38}$$

*and in this last case*

$$\langle N_1,...,N_n|T_m^{(K)}(\lambda)|t\rangle = t_m(\lambda)\langle N_1,...,N_n|t\rangle. \tag{A.39}$$

*Proof.* By the definitions of $N_m^\pm$ it is clear that for any SoV covector must hold the inequalities (A.37) and that the two identities in (A.38) are equivalent. Then under the condition $N_m^T = N$ the following identities hold:

$$N_{m-1}^- = 2N_{n-m} + N_{n+1-m} \neq 0, \quad N_{m-1}^+ = 0, \tag{A.40}$$

$$N_{m+1}^- = 0, \quad N_{m+1}^+ = N_{n-m} + 2N_{n+1-m} \neq 0, \tag{A.41}$$

so that using the standard interpolation formula and fusion identities we prove the above relation (A.39). $\qquad\square$

### A.2.2 Generation of loops

Let us observe that if, as assumed, $\bar{N}_m^- \neq 0$ and $\bar{N}_m^+ \neq 0$ then it holds also $N_{m-1}^- \neq 0$ and $N_{m+1}^+ \neq 0$ while it may hold $\bar{N}_{m-1}^+ = 0$ and $\bar{N}_{m+1}^- = 0$. If these last identities holds we have proven the identity (A.5) for the covector considered. Indeed, we have that the Laurent polynomial developments of both (A.17) and (A.18) coincide as the matrix elements (A.27) and (A.32), respectively, coincide with (A.28) and (A.33), as we can apply for them the identity (A.5) for $m-1$ and $m+1$ being $\bar{N}_{m-1}^+ = 0$ and $\bar{N}_{m+1}^- = 0$.

If $\bar{N}_{m-1}^+ \neq 0$ then the terms (A.27) and (A.28) have to be developed respectively by using the interpolation formula (A.22) and (A.23) with the choice of the points (A.24). So that, for all the $b$ such that in the covector $\langle N_1,...,N_n|$ of (A.27) it holds $n+1-m \leq h_b \leq n-1$, we are lead to the matrix elements:

$$\langle (N_m^- = \bar{N}_m^- - 1, N_m^+ \leq \bar{N}_m^+)|T_m^{(K)}(\xi_b)|t\rangle, \tag{A.42}$$

and

$$t_m(\xi_b)\langle (N_m^- = \bar{N}_m^- - 1, N_m^+ \leq \bar{N}_m^+)|t\rangle, \tag{A.43}$$

where[5] $N_m^+ = \bar{N}_m^+ + \delta_{n-m}^{h_a} - 1$. We have done in this way one loop (we call it $L_{-1,+1}^{(m)}$) to matrix elements containing $T_m^{(K)}(\lambda)$ and $t_m(\lambda)$ over new covectors, where we have reduced of one

---

[5]This value of $N_m^+$ in the second step covectors, appearing on the left of (A.42)-(A.43), is computed by using the line (A.30). We have to observe that the value of $N_m^+$ is now smaller of one unit of the one in the first step covectors, appearing on the left of (A.27)-(A.28). Moreover, we have to take in consideration that the value of $h_a$ in the first step covectors is bigger of one unit of the value in the original covector, appearing on the left of (A.25)-(A.26). Similar remarks apply for the values of $N_m^-$ after (A.48).

unit the value of the $N_m^-$ w.r.t. the original value $\bar{N}_m^-$ and reduced of at least one unit the total number:

$$N_m^- + N_m^+ \leq \bar{N}_m^- + \bar{N}_m^+ - 1. \tag{A.44}$$

However, for all the $b$ such that in the covector $\langle N_1, ..., N_n|$ of (A.27) it holds $0 \leq h_b \leq n-m$, we are lead to the matrix elements:

$$\langle (N_{m-2}^- = \bar{N}_{m-2}^- - 2, N_{m-2}^+ = \bar{N}_{m-2}^+)|T_{m-2}^{(K)}(\xi_b - \eta)|t\rangle, \tag{A.45}$$

and

$$t_{m-2}(\xi_b - \eta)\langle (N_{m-2}^- = \bar{N}_{m-2}^- - 2, N_{m-2}^+ = \bar{N}_{m-2}^+)|t\rangle. \tag{A.46}$$

Similarly, if $\bar{N}_{m+1}^- \neq 0$ then the terms (A.32) and (A.33) have to be developed by using respectively the interpolation formula (A.22) and (A.23) with the choice of the points (A.24). So that, for all the $b$ such that in the covector $\langle N_1, ..., N_n|$ of (A.32) it holds $0 \leq h_b \leq n-2-m$, we are lead to the matrix elements:

$$\langle (N_m^- \leq \bar{N}_m^-, N_m^+ = \bar{N}_m^+ - 1)|T_m^{(K)}(\xi_b - \eta)|t\rangle, \tag{A.47}$$

and

$$t_m(\xi_b - \eta)\langle (N_m^- \leq \bar{N}_m^-, N_m^+ = \bar{N}_m^+ - 1)|t\rangle, \tag{A.48}$$

where $N_m^- = \bar{N}_m^- + \delta_{n-m+1}^{h_a} - 1$. We have done in this way one loop (we call it $L_{+1,-1}^{(m)}$) to matrix elements containing $T_m^{(K)}(\lambda)$ and $t_m(\lambda)$ over new covectors where we have reduced of one unit the value of the $N_m^+$ w.r.t. the original value $\bar{N}_m^+$ and reduced of at least one unit the total number:

$$N_m^- + N_m^+ \leq \bar{N}_m^- + \bar{N}_m^+ - 1. \tag{A.49}$$

However, for all the $b$ such that in the covector $\langle N_1, ..., N_n|$ of (A.32) it holds $n-1-m \leq h_b \leq n-1$, we are lead to the matrix elements:

$$\langle (N_{m+2}^- = \bar{N}_{m+2}^-, N_{m+2}^+ = \bar{N}_{m+2}^+ - 2)|T_{m+2}^{(K)}(\xi_b)|t\rangle, \tag{A.50}$$

and

$$t_{m+2}(\xi_b)\langle (N_{m+2}^- = \bar{N}_{m+2}^-, N_{m+2}^+ = \bar{N}_{m+2}^+ - 2)|t\rangle. \tag{A.51}$$

Let us observe that from $\bar{N}_m^- \neq 0$ and $\bar{N}_m^+ \neq 0$ then it holds also $N_{m-2}^- = \bar{N}_{m-2}^- - 2 \neq 0$ and $N_{m+2}^+ = \bar{N}_{m+2}^+ - 2 \neq 0$ while it may hold that $\bar{N}_{m-2}^- = 0$ and $\bar{N}_{m+2}^- = 0$. If these last identities holds then the matrix elements containing $T_{m\pm2}^{(K)}(\lambda)$ and $t_{m\pm2}(\lambda)$ are known to coincide and we are left with the computation of those of $T_m^{(K)}(\lambda)$ and $t_m(\lambda)$ after one cycle $L_{1+,1-}^{(m)}$ and $L_{1-,1+}^{(m)}$. If instead $N_{m\mp2}^{\pm} \neq 0$, we have to repeat for $T_{m\mp2}^{(K)}(\lambda)$ and $t_{m\pm2}(\lambda)$ the same procedure developed for $T_m^{(K)}(\lambda)$ and $t_m(\lambda)$.

In the boundary cases, i.e. for $m = 1$ and $m = n-1$, we have that to compute

$$\langle \bar{N}_1, ..., \bar{N}_n|T_1^{(K)}(\lambda)|t\rangle \text{ and } \langle \bar{N}_1, ..., \bar{N}_n|T_{n-1}^{(K)}(\lambda)|t\rangle \tag{A.52}$$

and

$$t_1(\lambda)\langle \bar{N}_1, ..., \bar{N}_n|t\rangle \text{ and } t_{n-1}(\lambda)\langle \bar{N}_1, ..., \bar{N}_n|t\rangle, \tag{A.53}$$

for any $\lambda \in \mathbb{C}$, we have just to be able to compute matrix elements of the type:

$$\langle \bar{N}_1, ..., \bar{N}_{n-1}, \bar{N}_n|T_1^{(K)}(\xi_a - \eta)|t\rangle = \langle \bar{N}_1, ..., \bar{N}_{n-1} + 1, \bar{N}_n - 1|T_2^{(K)}(\xi_a)|t\rangle, \tag{A.54}$$

and

$$t_1(\xi_a - \eta)\langle \bar{N}_1, ..., \bar{N}_{n-1}, \bar{N}_n|t\rangle = t_2(\xi_a)\langle \bar{N}_1, ..., \bar{N}_{n-1} + 1, \bar{N}_n - 1|t\rangle, \tag{A.55}$$

for $h_a = n - 1$ and, respectively,

$$\langle \bar{N}_1, \bar{N}_2, ..., \bar{N}_n | T_{n-1}^{(K)}(\xi_a) | t \rangle = \langle \bar{N}_1 - 1, \bar{N}_2 + 1, ..., \bar{N}_n | T_{n-2}^{(K)}(\xi_a - \eta) | t \rangle, \qquad (A.56)$$

and

$$t_{n-1}(\xi_a) \langle \bar{N}_1, \bar{N}_2, ..., \bar{N}_n | t \rangle = t_{n-2}(\xi_a - \eta) \langle \bar{N}_1 - 1, \bar{N}_2 + 1, ..., \bar{N}_n | t \rangle, \qquad (A.57)$$

for $h_a = 0$. Here, for $\bar{N}_2^- = 0$ (for $\bar{N}_{n-2}^+ = 0$) the matrix elements containing $T_2^{(K)}(\xi_a)$ and $t_2(\xi_a)$ ($T_{n-2}^{(K)}(\xi_a - \eta)$ and $t_{n-2}(\xi_a - \eta)$) are known to coincide otherwise we have to expand them in the usual way by using the interpolation formula and this produces a loop $L_{+1,-1}^{(1)}$ ($L_{-1,+1}^{(n-1)}$).

Following several times the above procedure, of interpolation expansions and use of fusions, we can also generate $a$-steps loops of down or up types. Here we denote with $L_{-a,+a}^{(m)}$ the $a$-steps loop obtained as $a$ consecutive down and then $a$ consecutive up, and with $L_{+a,-a}^{(m)}$ the $a$-steps loop obtained as $a$ consecutive up and then $a$ consecutive down. In the following lemma we show that these produce the same minimal type of shifts of the one loop

*Lemma A.3. Let us assume that starting from the matrix elements (A.17) and (A.18) we develop a x-steps loop of down type $L_{-x,+x}^{(m)}$ then we are lead to matrix elements of the form:*

$$\langle (N_m^- \leq \bar{N}_m^- - 1, N_m^+ \leq \bar{N}_m^+) | T_m^{(K)}(\xi_a) | t \rangle, \qquad (A.58)$$

*and*

$$t_m(\xi_a) \langle (N_m^- \leq \bar{N}_m^- - 1, N_m^+ \leq \bar{N}_m^+) | t \rangle. \qquad (A.59)$$

*While if we develop an x-step loop of up type $L_{+x,-x}^{(m)}$ then we are lead to matrix elements of the form:*

$$\langle (N_m^- \leq \bar{N}_m^-, N_m^+ = \bar{N}_m^+ - 1) | T_m^{(K)}(\xi_a - \eta) | t \rangle, \qquad (A.60)$$

*and*

$$t_m(\xi_a - \eta) \langle (N_m^- \leq \bar{N}_m^-, N_m^+ = \bar{N}_m^+ - 1) | t \rangle, \qquad (A.61)$$

*so that in both the case it holds:*

$$N_m^- + N_m^+ \leq \bar{N}_m^- + \bar{N}_m^+ - 1. \qquad (A.62)$$

*Proof.* Let us assume that the original state $\langle \bar{N}_1, \bar{N}_2, ..., \bar{N}_n |$ is such that starting from the matrix elements

$$\langle \bar{N}_1, \bar{N}_2, ..., \bar{N}_n | T_m^{(K)}(\xi_a - \eta) | t \rangle, \qquad (A.63)$$

and

$$t_m(\xi_a - \eta) \langle \bar{N}_1, \bar{N}_2, ..., \bar{N}_n | t \rangle, \qquad (A.64)$$

we can implement a loop process of the type $L_{+x,-x}^{(m)}$, then the consecutive expansions up and down produce matrix elements associated to covectors with the following modifications on

the index:

$$N_n = \bar{N}_n - \sum_{b=0}^{x-1} \delta_{t_{(b,+)},n-1},$$

$$N_{n-r} = \bar{N}_{n-r} - \sum_{b=0}^{x-1}(\delta_{t_{(b,+)},n-r-1} - \delta_{t_{(b,+)},n-r}), \text{ for any } r \in \{1,...,m-1\},$$

$$N_{n-m} = \bar{N}_{n-m} - \sum_{b=1}^{x-1}(\delta_{t_{(b,+)},n-m-1} - \delta_{t_{(b,+)},n-m}) + \delta_{t_{(0,+)},n-m} + \delta_{t_{(1,-)},n-m-2},$$

$$N_{n-(m+r)} = \bar{N}_{n-(m+r)} - \sum_{b=1+r}^{x-1}(\delta_{t_{(b,+)},n-(m+r)-1} - \delta_{t_{(b,+)},n-(m+r)}) + \delta_{t_{(r,+)},n-(m+r)}$$

$$+ \delta_{t_{(r+1,-)},n-(m+r+2)} - \sum_{b=1}^{r}(\delta_{t_{(b,-)},n-(m+r+1)} - \delta_{t_{(b,-)},n-(m+r+2)}), \,\forall r \in \{1,...,x-2\},$$

$$N_{n-(m+x-1)} = \bar{N}_{n-(m+x-1)} + \delta_{t_{(r,+)},n-(m+x-1)} + \delta_{t_{(x,-)},n-(m+x+1)}$$

$$- \sum_{b=1}^{x-1}(\delta_{t_{(b,-)},n-(m+x)} - \delta_{t_{(b,-)},n-(m+x+1)}),$$

$$N_{n-(m+x+s)} = \bar{N}_{n-(m+x+s)} - \sum_{b=1}^{x}(\delta_{t_{(b,-)},n-(m+x+s+1)} - \delta_{t_{(b,-)},n-(m+x+s+2)}),$$

$$\forall s \in \{0,...,n-(m+x+2)\},$$

$$N_1 = \bar{N}_1 - \sum_{b=1}^{x} \delta_{t_{(b,-)},0}, \tag{A.65}$$

where we have used the following notations:

a) in the step $b+1$ up produced by fusion on matrix elements of the type:

$$\langle h_1^{(b,+)}, h_2^{(b,+)},...,h_N^{(b,+)}|T_{m+b}^{(K)}(\xi_a)|t\rangle, \qquad t_{m+b}(\xi_a)\langle h_1^{(b,+)}, h_2^{(b,+)},...,h_N^{(b,+)}|t\rangle, \tag{A.66}$$

then we have denoted $t_{(b,+)} = h_a^{(b,+)}$, for any $b \in \{0,...,x-1\}$.

b) in the step $b$ down produced by fusion on matrix elements of the type:

$$\langle h_1^{((x-1,+),(b,-))}, h_2^{((x-1,+),(b,-))},...,h_N^{((x-1,+),(b,-))}|T_{m+x+1-b}^{(K)}(\xi_a-\eta)|t\rangle,$$

$$t_{m+x+1-b}(\xi_a-\eta)\langle h_1^{((x-1,+),(b,-))}, h_2^{((x-1,+),(b,-))},...,h_N^{((x-1,+),(b,-))}|t\rangle, \tag{A.67}$$

then we have denoted $t_{(b,-)} = h_a^{((x-1,+),(b,-))}$, for any $b \in \{1,...,x\}$.

In writing the above contributions to the $N_i$ of the covector obtained by the full process $L_{+x,-x}^{(m)}$, we have considered that according to the expansion rule (A.24) and the fusion rule we can have an up/down step if and only if:

$$0 \le t_{(b,-)} \le n-1-(m+b), \quad \forall b \in \{1,...,x\},$$

$$n-(m+b) \le t_{(b,+)} \le n-1, \quad \forall b \in \{0,...,x-1\}. \tag{A.68}$$

Now from the above formulae it follows for the final state the following rules:

$$
\begin{aligned}
N_m^+ &= \bar{N}_m^+ - \sum_{s=n-m}^{n-1} \sum_{b=0}^{x-1} \delta_{t_{(b,+)},s} = \bar{N}_m^+ - 1 - \sum_{s=n-m}^{n-1} \sum_{b=1}^{x-1} \delta_{t_{(b,+)},s} \\
&\leq \bar{N}_m^+ - 1 \\
N_m^- &= \bar{N}_m^- - \sum_{s=0}^{n-m-2} \sum_{b=1}^{x} \delta_{t_{(b,-)},s} + \sum_{s=0}^{n-m} \sum_{b=0}^{x-1} \delta_{t_{(b,+)},s} \\
&= \bar{N}_m^- - x + \sum_{s=0}^{n-m} \sum_{b=0}^{x-1} \delta_{t_{(b,+)},s} \leq \bar{N}_m^-,
\end{aligned}
\tag{A.69, A.70}
$$

being:

$$
\sum_{s=n-m}^{n-1} \delta_{t_{(0,+)},s} = 1, \quad \sum_{s=0}^{n-m} \sum_{b=0}^{x-1} \delta_{t_{(b,+)},s} \leq x, \tag{A.71}
$$

$$
\sum_{s=0}^{n-m-2} \delta_{t_{(b,-)},s} = 1, \text{ for any } b \in \{1,...,x\}. \tag{A.72}
$$

$\square$

## A.3 Proof of Theorem 3.1

We have now the tools required to prove our Theorem 3.1

*Proof of Theorem 3.1: second part.* Let us use the above rules to prove the identities

$$
\langle \bar{N}_1,...,\bar{N}_n | T_1^{(K)}(\xi_a - \eta) | t \rangle = t_1(\xi_a - \eta) \langle \bar{N}_1,...,\bar{N}_n | t \rangle, \tag{A.73}
$$

for $h_a = n-1$, from which the identity (A.5) holds for $m=1$ and so for any $m \leq n-1$.

We proceed doing a first loop of type $L_{+,-}^{(1)}$ on both

$$
\langle \bar{N}_1,...,\bar{N}_n | T_1^{(K)}(\xi_a - \eta) | t \rangle \text{ and } t_1(\xi_a - \eta) \langle \bar{N}_1,...,\bar{N}_n | t \rangle, \tag{A.74}
$$

this leads to the same linear combination of $N$ matrix elements, those of loop type:

$$
\langle N_1^+ = \bar{N}_1^+ - 1, N_1^- \leq \bar{N}_1^- | T_1^{(K)}(\xi_b - \eta) | t \rangle \text{ and } t_1(\xi_b - \eta) \langle N_1^+ = \bar{N}_1^+ - 1, N_1^- \leq \bar{N}_1^- | t \rangle, \tag{A.75}
$$

and the others of the type:

$$
\langle N_3^+ = \bar{N}_3^+ - 2, N_3^- = \bar{N}_3^- | T_3^{(K)}(\xi_b) | t \rangle \text{ and } t_3(\xi_b) \langle N_3^+ = \bar{N}_3^+ - 2, N_3^- = \bar{N}_3^- | t \rangle. \tag{A.76}
$$

Now if $\bar{N}_1^+ = 1$, then the matrix elements in (A.75) coincides. If $\bar{N}_3^- = 0$, then the matrix elements in (A.76) coincides. If both these identities holds then all these $N$ matrix elements are proven to coincide which implies (A.73). Otherwise we make a second loop for the matrix elements in (A.76) if $\bar{N}_3^- \geq 1$. This leads to write each one of our original matrix elements (A.74) as the same linear combination of at most $N^3$ matrix elements, those of type (A.75) that we haven't further developed, new elements of the type[6] (A.75) generated by a 2 loop $L_{2+,2-}^{(1)}$, those of loop $L_{-,+}^{(3)}$ and $L_{+,-}^{(3)}$ which can be written in a common notation as it follows:

$$
\begin{aligned}
&\langle N_3^+ \leq \bar{N}_3^+ - 2 - r_3, N_3^- \leq \bar{N}_3^- - s_3 | T_3^{(K)}(\xi_b^{(h^{(3)})}) | t \rangle, \\
&t_3(\xi_b^{(h^{(3)})}) \langle N_3^+ \leq \bar{N}_3^+ - 2 - r_3, N_3^- \leq \bar{N}_3^- - s_3 | t \rangle,
\end{aligned}
\tag{A.77}
$$

---

[6]Indeed, as it is shown in the Lemma A.3, the rules for a loop with multiple steps up and down produce the same minimal type of shifts of the one loop.

with $h^{(3)} = 0$ if the loop is $L_{-,+}^{(3)}$ and $h^{(3)} = 1$ if the loop is $L_{+,-}^{(3)}$ with $r_3 + s_3 = 1$ and the others of the type:

$$\langle N_5^+ = \bar{N}_5^+ - 4, N_5^- = \bar{N}_5^- | T_5^{(K)}(\xi_b) | t \rangle \text{ and } t_5(\xi_b) \langle N_5^+ = \bar{N}_5^+ - 4, N_5^- = \bar{N}_5^- | t \rangle. \tag{A.78}$$

So after this process we are left just with the matrix elements of the type (A.75), (A.77) and (A.78).

Now we have to repeat the same type of considerations done in the first loop. In particular, for the matrix elements in (A.78) we do a further loop if the conditions $\bar{N}_5^- \geq 1$ is satisfied while for $\bar{N}_5^- = 0$, no further development is required as the matrix elements in (A.78) coincide. If this third loop is required this produces for each one of our original matrix elements (A.74) the development in the same linear combination of at most $N^5$ matrix elements, those of loop type (A.75) and (A.77), which we haven't developed, those of the type (A.77) which are generated by the development of (A.78) and correspond to a two steps up and down loop $L_{+2,-2}^{(3)}$, the one loop type $L_{-,+}^{(5)}$ and $L_{+,-}^{(5)}$ which can be written in a common notation as it follows:

$$\langle N_5^+ \leq \bar{N}_5^+ - 4 - r_5, N_5^- \leq \bar{N}_5^- - s_5 | T_3^{(K)}(\xi_b^{(h^{(5)})}) | t \rangle,$$
$$t_3(\xi_b^{(h^{(5)})}) \langle N_5^+ \leq \bar{N}_5^+ - 4 - r_5, N_5^- \leq \bar{N}_5^- - s_5 | t \rangle, \tag{A.79}$$

$h^{(5)} = 0$ if the loop is $L_{-,+}^{(5)}$ and $h^{(5)} = 1$ if the loop is $L_{+,-}^{(5)}$ with $r_5 + s_5 = 1$ and the new ones

$$\langle N_7^+ = \bar{N}_7^+ - 6, N_7^- = \bar{N}_7^- | T_7^{(K)}(\xi_b) | t \rangle \text{ and } t_7(\xi_b) \langle N_7^+ = \bar{N}_7^+ - 6, N_7^- = \bar{N}_7^- | t \rangle. \tag{A.80}$$

This process is continued up to $x$ loops, where $x$ is the smaller integer such that $\bar{N}_{1+2x}^- \geq 1$ and $\bar{N}_{1+2(x+1)}^- = 0$ or $x = \{(n-1)/2$ for $n$ odd, $(n-2)/2$ for $n$ even$\}$, in this way producing for each one of our original matrix elements (A.74) the development in the same linear combination of at most $N^{1+2x}$ matrix elements, of the type:

$$\langle N_{1+2a}^+ \leq \bar{N}_{1+2a}^+ - 2a - r_{1+2a}, N_{1+2a}^- \leq \bar{N}_{1+2a}^- - s_{1+2a} | T_{1+2a}^{(K)}(\xi_b^{(h^{(1+2a)})}) | t \rangle,$$
$$t_{1+2a}(\xi_b^{(h^{(1+2a)})}) \langle N_{1+2a}^+ \leq \bar{N}_{1+2a}^+ - 2a - r_{1+2a}, N_{1+2a}^- \leq \bar{N}_{1+2a}^- - s_{1+2a} | t \rangle, \tag{A.81}$$

with $r_{1+2a} + s_{1+2a} = 1$ for $1 \leq a \leq x - 1$ and the remaining one of the type:

$$\langle N_{1+2x}^+ = \bar{N}_{1+2x}^+ - 2x, N_{1+2x}^- = \bar{N}_{1+2x}^- | T_{1+2x}^{(K)}(\xi_b) | t \rangle,$$
$$t_{1+2x}(\xi_b) \langle N_{1+2x}^+ = \bar{N}_{1+2x}^+ - 2x, N_{1+2x}^- = \bar{N}_{1+2x}^- | t \rangle. \tag{A.82}$$

From this point any further loop development of (A.82) does not generate new type of terms but instead it generates matrix elements of the type:

$$\langle N_{2x-1}^+ \leq \bar{N}_{2x-1}^+ + 1 - 2x - r_{2x-1}, N_{2x-1}^- \leq \bar{N}_{2x-1}^- - s_{2x-1} | T_{2x-1}^{(K)}(\xi_b^{(h^{(2x-1)})}) | t \rangle,$$
$$t_{2x-1}(\xi_b^{(h^{(2x-1)})}) \langle N_{2x-1}^+ \leq \bar{N}_{2x-1}^+ + 1 - 2x - r_{2x-1}, N_{2x-1}^- \leq \bar{N}_{2x-1}^- - s_{2x-1} | t \rangle, \tag{A.83}$$

and

$$\langle N_{1+2x}^+ \leq \bar{N}_{1+2x}^+ - 2x, N_{1+2x}^- = \bar{N}_{1+2x}^- - 1 | T_{1+2x}^{(K)}(\xi_b) | t \rangle,$$
$$t_{1+2x}(\xi_b) \langle N_{1+2x}^+ \leq \bar{N}_{1+2x}^+ - 2x, N_{1+2x}^- = \bar{N}_{1+2x}^- - 1 | t \rangle. \tag{A.84}$$

Starting from the terms (A.82) making at most $\bar{N}^-_{1+2(x+1)}$ loops we arrive at matrix elements of the type (A.83) and to

$$\langle N^+_{1+2x} \le \bar{N}^+_{1+2x} - 2x, N^-_{1+2x} = 0|T^{(K)}_{1+2x}(\xi_b)|t\rangle,$$

$$t_{1+2x}(\xi_b)\langle N^+_{1+2x} \le \bar{N}^+_{1+2x} - 2x, N^-_{1+2x} = 0|t\rangle,$$

(A.85)

and so for them the coincidence of the matrix elements in (A.85) is known and we are left only with matrix elements of the type (A.81) for $1 \le a \le x-1$. Now we make a one loop developments for the matrix elements (A.81) with (A.81) for $a = x-1$, this produces matrix elements of the type (A.81) with $a = x-2$, loops term of the type

$$\langle N^T_{1+2(x-1)} \le \bar{N}^T_{1+2(x-1)} - 2x - 1|T^{(K)}_{1+2(x-1)}(\xi_b)|t\rangle,$$

(A.86)

$$t_{1+2(x-1)}(\xi_b)\langle N^T_{1+2(x-1)} \le \bar{N}^T_{1+2(x-1)} - 2x - 1|t\rangle$$

(A.87)

and new border terms of the type (A.84) with $N^-_{1+2x} \le \bar{N}^-_{1+2x} - 1$, either this integer is zero or we repeat for these border terms the same procedure explained above ending up with only matrix elements of the type (A.81) for $1 \le a \le x-2$ while for $a = x-1$, we are left with all terms of the type (A.87) so that we have reduced of one unit the value of $N^T_{1+2(x-1)}$ with respect to the value in (A.83) for which it was holding

$$N^T_{1+2(x-1)} \le \bar{N}^+_{2x-1} + 1 - 2x - r_{2x-1} + \bar{N}^-_{2x-1} - s_{2x-1} = \bar{N}^T_{2x-1} - 2x.$$

(A.88)

We can repeat this procedure now and after at most a total of $\bar{N}^T_{2x-1} - 2x - N$, we generate matrix elements with both $a = x-1$ and $a = x$ for which the identity between operator and scalar matrix elements is known. So on we are lead to develop matrix elements with $a = x-2$ reducing, at any step of the above procedure, at least of one unit the value of $N^T_{1+2(x-2)}$ up to arrive to matrix elements for which the identity of the operator and scalar terms is known. At this point we are left with $a = x-3$ and so on up to arrive to be left only with matrix elements of the type (A.81) for $a = 1$ for which we have $N^+_1 \le \bar{N}^+_1 - 1$, here if we want we can do induction and prove the theorem.

However, it is interesting to get a bound on the maximal number of loops to implement in order to be reduced to linear combinations of matrix elements for which the identity (A.5) applies. From the above analysis, to get the reduction of at least one unit in $N^+_1$, we have to implement before $x$ loops, to generate all the types of allowed matrix elements, then we have to do less than

$$N^-_{1+2x} \prod_{a=1}^{x-1} (\bar{N}^T_{2a+1} - (2a+1) - N),$$

(A.89)

loops. This means that using the above procedure, we have to do less than

$$\bar{N}_T \equiv \bar{N}^+_1 (x + N^-_{1+2x} \prod_{a=1}^{x-1} (\bar{N}^T_{2a+1} - (2a+1) - N)),$$

(A.90)

loops in order to generate only matrix elements for which the identity (A.5) applies. This means that starting from the original matrix elements (A.17) and (A.18), by implementing all these loops, they will be rewritten as the linear combination of less than $N^{1+2\bar{N}_T}$ matrix elements for which the identity (A.5) applies with the same coefficients, namely for any $i = 1, \ldots, N$ we have:

$$\langle \bar{N}_1, ..., \bar{N}_n|T^{(K)}_1(\xi_i - \eta)|t\rangle = \sum_{r=1}^{n-1}\sum_{a=1}^{N}\sum_{h=0}^{1} \sum_{\substack{\{N_1, ..., N_n\} \in \\ S^{(m,\{\bar{N}\})}_{r,a,h}}} C^{(1,\{\bar{N}\},r,a,h,i)}_{\{N_1,...,N_n\}} \langle N_1, ..., N_n|T^{(K)}_r(\xi^{(h)}_a)|t\rangle, \quad (A.91)$$

and

$$t_1(\xi_i - \eta)\langle \bar{N}_1, ..., \bar{N}_n | t \rangle = \sum_{r=1}^{n-1} \sum_{a=1}^{N} \sum_{h=0}^{1} \sum_{\substack{\{N_1,...,N_n\} \in \\ S_{r,a,h}^{(m,\{\bar{N}\})}}} C_{\{N_1,...,N_n\}}^{(1,\{\bar{N}\},r,a,h,i)} t_r(\xi_a^{(h)}) \langle N_1, ..., N_n | t \rangle, \quad \text{(A.92)}$$

from which the proof of the Theorem directly follows. □

It is interesting to remark here that all the formulae derived in this Appendix could have been obtained exactly in the same manner for the products of operators themselves appearing in any of the considered average value between the co-vector $\langle S |$ and the vector $| t \rangle$. Hence a direct consequence of the above relation is the equation (3.15) with the property that the coefficients $C_{\mathbf{hh}'}^{(a)}$ are obtained from the fusion relations and asymptotic behavior of the transfer matrices.

# B  Appendix B

This appendix is devoted to complete the proof of the Theorem 4.1. In a first subsection, we first recall for self-consistence some elementary properties of the algebraic functions that we will use. Then we present in the second subsection the proof the Theorem 4.1.

## B.1  Some elementary properties of algebraic functions

Let us present a couple of elementary lemmas on algebraic functions. First, let us recall that a function $f(x_1, ..., x_N)$ of $N$ variables:

$$f : (x_1, ..., x_N) \in \mathbb{C}^N \rightarrow f(x_1, ..., x_N) \in \mathbb{C}, \quad \text{(B.1)}$$

is by definition algebraic iff there exist $M+1$ polynomials $a_m(x_1, ..., x_N)$ such that it holds:

$$P(y | x_1, ..., x_N) = 0 \text{ for } y \equiv f(x_1, ..., x_N), \quad \text{(B.2)}$$

where we have defined the polynomial $P(y | x_1, ..., x_N)$ as it follows:

$$P(y | x_1, ..., x_N) = \sum_{m=0}^{M} y^m a_m(x_1, ..., x_N). \quad \text{(B.3)}$$

Moreover, we say that $f(x_1, ..., x_N)$ is an algebraic function of order $M$ if the polynomial $a_M(x_1, ..., x_N)$ is not identically zero. Clearly, to any set of polynomial $a_m(x_1, ..., x_N)$ for $m \in \{0, ..., M\}$, under the condition $a_M(x_1, ..., x_N)$ not identically zero, are associated $M$ algebraic functions $p_i(x_1, ..., x_N)$ of order $M$ such that:

$$P(y | x_1, ..., x_N) = a_M(x_1, ..., x_N) \prod_{m=1}^{M} (y - p_m(x_1, ..., x_N)). \quad \text{(B.4)}$$

*Lemma* B.1. *Let us fix $P(y | x_1, ..., x_N)$ a degree $M$ polynomial in $y$ of the form (B.3), then the $M$ associated algebraic functions $p_m(x_1, ..., x_N)$ are all nonzero almost for any $(x_1, ..., x_N) \in \mathbb{C}^N$ if (and only if) there exists at least a point $(\bar{x}_1, ..., \bar{x}_N) \in \mathbb{C}^N$ such that:*

$$p_m(\bar{x}_1, ..., \bar{x}_N) \neq 0 \quad \forall m \in \{1, ..., M\}. \quad \text{(B.5)}$$

*Proof.* The proof is an immediate consequence of the following identity:

$$a_0(x_1, ..., x_N) = (-1)^M a_M(x_1, ..., x_N) \prod_{m=1}^{M} p_m(x_1, ..., x_N), \qquad (B.6)$$

of the fact that by assumption $a_0(x_1, ..., x_N)$ and $a_M(x_1, ..., x_N)$ are polynomials and $a_M(x_1, ..., x_N)$ is not identically zero. Indeed, the inequality (B.5) implies that it holds:

$$a_0(\bar{x}_1, ..., \bar{x}_N) \neq 0, \qquad (B.7)$$

so that, being it a polynomial, it is not identically zero. So the product

$$\prod_{m=1}^{M} p_m(x_1, ..., x_N) \neq 0, \qquad (B.8)$$

for almost any $(x_1, ..., x_N) \in \mathbb{C}^N$ and the same has to be true for any one of the algebraic functions $p_m(x_1, ..., x_N)$. $\qquad \square$

It is interesting to point out that any algebraic function $f(x_1, ..., x_N)$ of degree $M$ can be naturally interpreted as the eigenvalue of a $N$-parameters polynomial family of $M \times M$ square matrices. Indeed, let us denote with $P(y|x_1, ..., x_N)$ a degree $M$ polynomial in $y$ of the form (B.3) such that $f(x_1, ..., x_N)$ is one of its roots (B.1). Then, $f(x_1, ..., x_N)$ is an eigenvalue of a family of square matrices $\mathsf{P}(x_1, ..., x_N)$ with characteristic polynomial coinciding with $P(y|x_1, ..., x_N)$. Note that we have the following representation of $\mathsf{P}(x_1, ..., x_N)$ in terms of the Frobenius companion matrix:

$$\mathsf{P}(x_1, ..., x_N) = V_M \mathsf{C}(x_1, ..., x_N) V_M^{-1}, \qquad (B.9)$$

where $V_M$ is any invertible $M \times M$ square matrix and

$$\mathsf{C}(x_1, ..., x_N) = \begin{pmatrix} 0 & 1 & 0 & \cdots & 0 \\ 0 & 0 & 1 & \ddots & \vdots \\ \vdots & \vdots & \ddots & \ddots & \vdots \\ 0 & 0 & \cdots & 0 & 1 \\ -\mathsf{a}_0(\{x_i\}) & -\mathsf{a}_1(\{x_i\}) & \cdots & -\mathsf{a}_{M-2}(\{x_i\}) & -\mathsf{a}_{M-1}(\{x_i\}) \end{pmatrix}_{M \times M}, \qquad (B.10)$$

and $\mathsf{a}_j(x_1, ..., x_N)$ are the following rational functions:

$$\mathsf{a}_j(x_1, ..., x_N) = a_j(x_1, ..., x_N)/a_M(x_1, ..., x_N). \qquad (B.11)$$

Indeed, the characteristic polynomial associated to $\mathsf{P}(x_1, ..., x_N)$ is

$$-P(y|x_1, ..., x_N)/a_M(x_1, ..., x_N) = \prod_{m=1}^{M} (y - p_m(x_1, ..., x_N)), \qquad (B.12)$$

which has the same algebraic function roots of $P(y|x_1, ..., x_N)$. This observation allows to prove easily the following:

*Lemma* B.2. *Let $p(x_1, ..., x_N)$ and $q(x_1, ..., x_N)$ be any couple of algebraic functions of order $M$ and $R$, respectively, then their sum and product are as well algebraic functions now of order at most $M + R$.*

*Proof.* Let us denote with $P(y|x_1,...,x_N)$ a degree $M$ polynomial in $y$ of the form (B.3) of which $p(x_1,...,x_N)$ is an algebraic function root. Similarly, let us denote with $Q(y|x_1,...,x_N)$ a degree $R$ polynomial in $y$ of the form

$$Q(y|x_1,...,x_N) = \sum_{r=0}^{R} y^r b_r(x_1,...,x_N) = b_R(x_1,...,x_N) \prod_{r=1}^{R} (y - q_r(x_1,...,x_N)), \qquad \text{(B.13)}$$

where $b_m(x_1,...,x_N)$ are polynomials and $b_R(x_1,...,x_N)$ is not identically zero, of which $q(x_1,...,x_N)$ is an algebraic function root. Then, let us denote with $C_P(x_1,...,x_N)$ and $C_Q(x_1,...,x_N)$ the $M \times M$ and $R \times R$ families of Frobenius companion matrices associated respectively to the polynomials $P(y|x_1,...,x_N)$ and $Q(y|x_1,...,x_N)$ so that $p(x_1,...,x_N)$ and $q(x_1,...,x_N)$ are eigenvalues of the matrices $C_P(x_1,...,x_N)$ and $C_Q(x_1,...,x_N)$, respectively. Let us introduce the matrices:

$$A_{p+q}(x_1,...,x_N) = V_M C_P(x_1,...,x_N) V_M^{-1} \otimes I_R + I_M \otimes V_R C_Q(x_1,...,x_N) V_R^{-1}, \qquad \text{(B.14)}$$
$$A_{p*q}(x_1,...,x_N) = V_M C_P(x_1,...,x_N) V_M^{-1} \otimes V_R C_Q(x_1,...,x_N) V_R^{-1}, \qquad \text{(B.15)}$$

where $V_M$ and $V_R$ are any invertible $M \times M$ and $R \times R$ square matrix, while we have denoted with $I_M$ and $I_R$ the identity $M \times M$ and $R \times R$ square matrices. Then, denoted with $\Sigma_{A_{p+q}(x_1,...,x_N)}$ and $\Sigma_{A_{p*q}(x_1,...,x_N)}$ the set of the eigenvalues of $A_{p+q}(x_1,...,x_N)$ and $A_{p*q}(x_1,...,x_N)$, our statement is a direct consequence of the following spectral properties:

$$\Sigma_{A_{p+q}(x_1,...,x_N)} = \{f(x_1,...,x_N) : f(x_1,...,x_N) = p_m(x_1,...,x_N) + q_r(x_1,...,x_N)\}, \qquad \text{(B.16)}$$
$$\Sigma_{A_{p*q}(x_1,...,x_N)} = \{f(x_1,...,x_N) : f(x_1,...,x_N) = p_m(x_1,...,x_N) q_r(x_1,...,x_N)\}. \qquad \text{(B.17)}$$

for all the $(r,m) \in \{1,...,M\} \times \{1,...,R\}$. $\qquad\square$

The previous lemma implies the following

*Corollary* B.1. *Let us take a polynomial of the form*

$$T(y_1,...,y_S|x_1,...,x_N) = \sum_{r_1,...,r_S=0}^{R} \prod_{a=1}^{S} y_a^{r_a} \alpha_r(x_1,...,x_N), \qquad \text{(B.18)}$$

*and $S$ algebraic functions $f_m(x_1,...,x_N)$, then*

$$t(x_1,...,x_N) \equiv T(f_1(x_1,...,x_N),...,f_S(x_1,...,x_N)|x_1,...,x_N) \qquad \text{(B.19)}$$

*is itself an algebraic function.*

## B.2 Proof of Theorem 4.1

The proof of the Theorem 4.1 is now developed using as main ingredient the properties of the algebraic functions.

*Proof of Theorem 4.1.* Let us now come to the proof of the existence of the polynomial $\varphi_t(\lambda)$ of maximal degree $N$ satisfying the TQ equation, once we assume that $t_1(\lambda)$ is a transfer matrix eigenvalue. We have to satisfy the system:

$$t_1(\xi_b) \varphi_t^{(i)}(\xi_b) = k_i a(\xi_b) \varphi_t^{(i)}(\xi_b - \eta), \qquad \forall b \in \{1,...,N\}, i \in \{1,...,n\}, \qquad \text{(B.20)}$$

and the conditions

$$\varphi_t^{(i)}(\xi_b) \neq 0 \quad, \forall b \in \{1,...,N\}. \qquad \text{(B.21)}$$

Moreover, saying that $\varphi_t^{(i)}(\lambda)$ is a polynomial of maximal degree $N$ is equivalent to say that $\varphi_t^{(i)}(\lambda)$ can be written in the following form:

$$\varphi_t^{(i)}(\lambda) = \sum_{a=1}^{N+1} \prod_{\substack{b=1 \\ b \neq a}}^{N+1} \frac{\lambda - \xi_b}{\xi_a - \xi_b} \, \varphi_t^{(i)}(\xi_a). \tag{B.22}$$

In (B.22), $\xi_{N+1}$ is an arbitrary complex number, different from $\xi_1, \dots, \xi_N$, which can be chosen at our convenience. Hence the system (B.20) is equivalent to a homogeneous linear system of $N$ equations for the $N+1$ unknowns $\varphi_t^{(i)}(\xi_1), \dots, \varphi_t^{(i)}(\xi_{N+1})$, which can alternatively be thought of as an inhomogeneous linear system for the $N$ unknowns $\varphi_t^{(i)}(\xi_1), \dots, \varphi_t^{(i)}(\xi_N)$ in terms of the $(N+1)$-th one $\varphi_t^{(i)}(\xi_{N+1})$:

$$\sum_{b=1}^{N} [C_{i,\xi_{N+1}}^{(t)}]_{ab} \, \varphi_t^{(i)}(\xi_b) = -\prod_{\ell=1}^{N} \frac{\xi_a - \xi_\ell - \eta}{\xi_{N+1} - \xi_\ell} \, \varphi_t^{(i)}(\xi_{N+1}). \tag{B.23}$$

The elements of the matrix $C_{1,\xi_{N+1}}^{(t)}$ of this linear system are

$$[C_{i,\xi_{N+1}}^{(t)}]_{rs} = -\delta_{rs} \frac{t_1(\xi_r)}{k_i a(\xi_r)} + \prod_{\substack{c=1 \\ c \neq s}}^{N+1} \frac{\xi_r - \xi_c - \eta}{\xi_s - \xi_c} \qquad \forall r, s \in \{1, \dots, N\}. \tag{B.24}$$

If we can prove that $\det_N[C_{i,\xi_{N+1}}^{(t)}]$ is nonzero and finite for any $\xi_{N+1}$, up to a finite number of values, then, for any given choice of $\varphi_t^{(i)}(\xi_{N+1}) \neq 0$, there exists one and only one nontrivial solution $(\varphi_t^{(i)}(\xi_1), \dots, \varphi_t^{(i)}(\xi_N))$ of the system (B.23), which is given by Cramer's rule:

$$\varphi_t^{(i)}(\xi_j) = \varphi_t^{(i)}(\xi_{N+1}) \frac{\det_N[C_{i,\xi_{N+1}}^{(t|j)}]}{\det_N[C_{i,\xi_{N+1}}^{(t)}]}, \qquad \forall j \in \{1, \dots, N\}, \tag{B.25}$$

with matrices $C_{i,\xi_{N+1}}^{(t|j)}$ defined as

$$[C_{i,\xi_{N+1}}^{(t|j)}]_{rs} = (1 - \delta_{s,j})[C_{i,\xi_{N+1}}^{(t)}]_{rs} - \delta_{s,j} \prod_{\ell=1}^{N} \frac{\xi_r - \xi_\ell - \eta}{\xi_{N+1} - \xi_\ell}, \tag{B.26}$$

for all $r, s \in \{1, \dots, N\}$. Then our statement is a consequence of the following proposition which ensures that all these determinants are nonzero for almost any values of their parameters.

Now by using the rank one $N \times N$ matrix $\Delta_{\xi_{N+1}}(\lambda)$, defined in (4.23), the interpolation formula for $\varphi_t^{(i)}(\lambda)$ can be rewritten in a one determinant form:

$$\varphi_t^{(i)}(\lambda) = \frac{\det_N[C_{i,\xi_{N+1}}^{(t)} + \Delta_{\xi_{N+1}}(\lambda)]}{\det_N[C_{i,\xi_{N+1}}^{(T_1^{(K)})}]} \prod_{c=1}^{N} \frac{\lambda - \xi_c}{\xi_{N+1} - \xi_c}, \tag{B.27}$$

or equivalently:

$$\varphi_t^{(i)}(\lambda) = \frac{\det_N[C_{i,\xi_{N+1}}^{(t)} + \bar{\Delta}_{\xi_{N+1}}(\lambda)]}{\det_N[C_{i,\xi_{N+1}}^{(T_1^{(K)})}]} + \prod_{c=1}^{N} \frac{\lambda - \xi_c}{\xi_{N+1} - \xi_c} - 1, \tag{B.28}$$

where we have defined:

$$[\bar{\Delta}_{\xi_{N+1}}(\lambda)]_{ab} = -\prod_{c=1, c \neq b}^{N+1} \frac{\lambda - \xi_c}{\xi_b - \xi_c} \prod_{c=1}^{N} \frac{\xi_a - \xi_c - \eta}{\xi_{N+1} - \xi_c} \qquad \forall a, b \in \{1, \dots, N\}. \tag{B.29}$$

$\square$

Let us define the functions:

$$f_t(\xi_{N+1}, \{\xi_{j\leq N}\}, \{k_i\}, \eta) = \prod_{a=1}^{N} \prod_{\substack{b=1 \\ b\neq a}}^{N+1} (\xi_a - \xi_b) \det_N \left[ C_{i,\xi_{N+1}}^{(t)} \right]_{rs},$$
(B.30)

$$g_{t,j}(\xi_{N+1}, \{\xi_{j\leq N}\}, \{k_i\}, \eta) = \prod_{a=1}^{N} \prod_{\substack{b=1 \\ b\neq a}}^{N+1} (\xi_a - \xi_b) \det_N \left[ C_{i,\xi_{N+1}}^{(t|j)} \right]_{rs},$$
(B.31)

where the $\{k_1, ..., k_n\}$ are the eigenvalues of the twist matrix $K$, then the following proposition holds:

*Proposition* B.1. *Let* $t_l(\lambda)$ *be the generic transfer matrix eigenvalue, with* $l \in \{1, \ldots, n^N\}$, *then* $f_{t_l}(\xi_{N+1}, \{\xi_{j\leq N}\}, \{k_i\})$ *and* $g_{t_l,j}(\xi_{N+1}, \{\xi_{j\leq N}\}, \{k_i\})$, *for any* $j \in \{1, \ldots, N\}$, *are polynomial in* $\xi_{N+1}$ *which for any value of* $\xi_{N+1}$, *up to a finite number of values, are nonzero for almost any values of* $\{\xi_1, \ldots, \xi_N\}$, $\{k_i\}$ *and* $\eta$.

*Proof.* First of all we have to remark that the transfer matrix of the fundamental representation of $Y(gl_n)$ associated to a twist matrix with eigenvalues $\{k_{i\leq n}\}$ is a polynomial of degree $N$ in $\lambda$ and $\eta$, is a polynomial of degree 1 in the $\{k_{i\leq n}\}$ and in the inhomogeneities $\{\xi_1, \ldots, \xi_N\}$. Then, the transfer matrix eigenvalues $t(\lambda | \{\xi_1, \ldots, \xi_N\}, \{k_i\}, \eta)$ are degree $N$ polynomials in $\lambda$ and, for any fixed value of $\lambda$, they are algebraic functions of maximal order $n^N$ in the variables $\{\xi_1, \ldots, \xi_N\}$, $\{k_i\}$ and $\eta$. Indeed, they are the zeros of the characteristic polynomial of the one parameter family of commuting transfer matrices of the following form:

$$P_T(y | \lambda, \{\xi_1, \ldots, \xi_N\}, \{k_i\}, \eta) = \sum_{b=0}^{n^N} y^b a_b(\lambda, \{\xi_1, \ldots, \xi_N\}, \{k_i\}, \eta)$$
(B.32)

$$= a_{n^N}(\lambda, \{\xi_1, \ldots, \xi_N\}, \{k_i\}, \eta)$$

$$\times \prod_{b=1}^{n^N} (y - t_b(\lambda | \{\xi_1, \ldots, \xi_N\}, \{k_i\}, \eta)), \quad \text{(B.33)}$$

where the $a_b(\lambda, \{\xi_{j\leq N}\}, \{k_i\}, \eta)$ are polynomials in their variables. From the previous corollary it follows that the functions $f_{t_l}(\xi_{N+1}, \{\xi_{j\leq N}\}, \{k_i\}, \eta)$ and $g_{t_l,j}(\xi_{N+1}, \{\xi_{j\leq N}\}, \{k_i\}, \eta)$, for any $j \in \{1, \ldots, N\}$, are polynomials in the $\xi_{N+1}$ and algebraic functions of their arguments $\{\xi_{j\leq N}\}$, $\{k_i\}$ and $\eta$, for any choice of the transfer matrix eigenvalue $t_l(\lambda | \{\xi_1, \ldots, \xi_N\}, \{k_i\}, \eta)$, i.e. for any $l \in \{1, \ldots, n^N\}$.

Let us introduce the new functions:

$$f_{y_1, \ldots, y_N}(\xi_{N+1}, \eta) = \prod_{a=1}^{N} \prod_{\substack{b=1 \\ b\neq a}}^{N+1} (\xi_a - \xi_b) \det_N [c_{y_1, \ldots, y_N}(\xi_{N+1})],$$
(B.34)

$$g_{y_1, \ldots, y_N, j}(\xi_{N+1}, \eta) = \prod_{a=1}^{N} \prod_{\substack{b=1 \\ b\neq a}}^{N+1} (\xi_a - \xi_b) \det_N [c_{y_1, \ldots, y_N}^{(j)}(\xi_{N+1})],$$
(B.35)

where we have defined:

$$[c_{y_1, \ldots, y_N}(\xi_{N+1})]_{ab} = -\delta_{ab} \frac{y_a}{k_1 a(\xi_a)} + \prod_{\substack{c=1 \\ c\neq b}}^{N+1} \frac{\xi_a - \xi_c - \eta}{\xi_b - \xi_c} \qquad \forall a, b \in \{1, \ldots, N\},$$
(B.36)

$$[c_{y_1, \ldots, y_N}^{(j)}(\xi_{N+1})]_{ab} = (1 - \delta_{b,j})[c_{y_1, \ldots, y_N}(\xi_{N+1})]_{ab} - \delta_{b,j} \prod_{\ell=1}^{N} \frac{\xi_a - \xi_\ell - \eta}{\xi_{N+1} - \xi_\ell}.$$
(B.37)

Clearly, it holds:

$$f_{t_l}(\xi_{N+1}, \{\xi_{j\leq N}\}, \{k_i\}, \eta) = f_{t_l(\xi_1),...,t_l(\xi_N)}(\xi_{N+1}, \eta), \tag{B.38}$$

$$g_{t_l,j}(\xi_{N+1}, \{\xi_{j\leq N}\}, \{k_i\}, \eta) = g_{t_l(\xi_1),...,t_l(\xi_N),j}(\xi_{N+1}, \eta), \tag{B.39}$$

moreover, it is clear that the function $f_{t_l}(\xi_{N+1}, \{\xi_{j\leq N}\}, \{k_i\}, \eta)$ and $g_{t_l,j}(\xi_{N+1}, \{\xi_{j\leq N}\}, \{k_i\}, \eta)$, for any $j \in \{1,...,N\}$, are algebraic functions of maximal order $n^{N^2}$ in their arguments $\{\xi_{j\leq N}\}$, $\{k_i\}$ and $\eta$. Indeed, defined:

$$\hat{P}_f(y|\xi_{N+1}, \{\xi_{j\leq N}\}, \{k_i\}, \eta) = \prod_{l_1=1}^{n^N} \cdots \prod_{l_N=1}^{n^N}(y - f_{t_{l_1}(\xi_1),...,t_{l_N}(\xi_N)}(\xi_{N+1}, \eta)), \tag{B.40}$$

$$\hat{P}_{g,j}(y|\xi_{N+1}, \{\xi_{j\leq N}\}, \{k_i\}, \eta) = \prod_{l_1=1}^{n^N} \cdots \prod_{l_N=1}^{n^N}(y - g_{t_{l_1}(\xi_1),...,t_{l_N}(\xi_N),j}(\xi_{N+1}, \eta)), \tag{B.41}$$

for any fixed $\xi_{N+1}$ these are single value algebraic functions of their arguments $\{\xi_{j\leq N}\}$, $\{k_i\}$ and $\eta$ so that there exist nonzero polynomials:

$$a_{n^{2N}}(\xi_{N+1}, \{\xi_{j\leq N}\}, \{k_i\}, \eta) \text{ and } a_{n^{2N}}^{(j)}(\xi_{N+1}, \{\xi_{j\leq N}\}, \{k_i\}, \eta), \tag{B.42}$$

such that defined

$$P_f(y|\xi_{N+1}, \{\xi_{j\leq N}\}, \{k_i\}, \eta) = a_{n^{2N}}(\xi_{N+1}, \{\xi_{j\leq N}\}, \{k_i\}, \eta)\hat{P}_f(y|\xi_{N+1}, \{\xi_{j\leq N}\}, \{k_i\}, \eta), \tag{B.43}$$

$$P_{g,j}(y|\xi_{N+1}, \{\xi_{j\leq N}\}, \{k_i\}, \eta) = a_{n^{2N}}^{(j)}(\xi_{N+1}, \{\xi_{j\leq N}\}, \{k_i\}, \eta)\hat{P}_{g,j}(y|\xi_{N+1}, \{\xi_{j\leq N}\}, \{k_i\}, \eta), \tag{B.44}$$

they are polynomials in all their variables and the $f_{t_{l_1}(\xi_1),...,t_{l_N}(\xi_N)}(\xi_{N+1}, \eta)$ and $g_{t_{l_1}(\xi_1),...,t_{l_N}(\xi_N),j}(\xi_{N+1}, \eta)$ are the respective associated algebraic roots. Let us remark now that the transfer matrix of the fundamental representation of $Y(gl_n)$ associated to a twist matrix satisfying the eigenvalues conditions $k_i = \delta_{1,i}$ is similar to the diagonal entry $A_1(\lambda)$ of the monodromy matrix. The spectrum of which is known and has the following form

$$\prod_{a=1}^{R}(\lambda - \xi_{\pi(a)}) \prod_{a=1+R}^{N}(\lambda - \xi_{\pi(a)} + \eta), \tag{B.45}$$

over all the values of $R \leq N$ and $\pi \in S_N$ (permutations of $\{1,...,N\}$). So that for any fixed $t_{l\leq n^N}(\lambda|\{k_i\})$, eigenvalue of the transfer matrix, there exist a $R_l \leq N$ and a $\pi_l \in S_N$ such that:

$$\lim_{k_i \to \delta_{1,i}} t_l(\lambda|\{k_i\}) = \prod_{a=1}^{R_l}(\lambda - \xi_{\pi_l(a)}) \prod_{a=1+R_l}^{N}(\lambda - \xi_{\pi_l(a)} + \eta). \tag{B.46}$$

It is then simple to observe that

$$f_{t_{l_1}(\xi_1),...,t_{l_N}(\xi_N)}(\xi_{N+1}, \eta)\Big|_{k_i=\delta_{1,i}} \prod_{l=1}^{N} a(\xi_l), \tag{B.47}$$

$$g_{t_{l_1}(\xi_1),...,t_{l_N}(\xi_N),j}(\xi_{N+1}, \eta)\Big|_{k_i=\delta_{1,i}} \prod_{l=1}^{N} a(\xi_l), \tag{B.48}$$

are polynomial of their parameters: $\xi_{N+1}$, $\{\xi_{j\leq N}\}$ and $\eta$. So that if we prove that they are nonzero in a point they are so almost everywhere. From the equation (B.46) it follows that for any $\{l_1,...,l_N\} \in \{1,...,n^N\}^{\otimes N}$ there exist a $R_{\{l_j\}} \leq N$ and a $\pi_{\{l_j\}} \in S_N$ such that:

$$t_{l_{\pi_{\{l_j\}}(i)}}(\xi_{\pi_{\{l_j\}}(i)}|\{\xi_{j\leq N}\}, \{k_i = \delta_{1,i}\}, \eta) = 0 \quad \text{if } i \leq R_{\{l_j\}} \tag{B.49}$$

$$t_{l_{\pi_{\{l_j\}}(i)}}(\xi_{\pi_{\{l_j\}}(i)}|\{\xi_{j\leq N}\}, \{k_i = \delta_{1,i}\}, \eta) \neq 0 \quad \text{if } R_{\{l_j\}} + 1 \leq i. \tag{B.50}$$

Then to compute

$$f_{t_{l_1}(\xi_1),\dots,t_{l_N}(\xi_N)}(\xi_{N+1},\eta)\Big|_{k_i=\delta_{1,i}}, \tag{B.51}$$

we define:

$$\tilde{\xi}^{(\pi)}_{\pi(a+1)} = \xi_{N+1} - (N-a)\eta, \qquad \forall a \in \{R_{\{l_j\}},\dots,N-1\}, \tag{B.52}$$

while the $\tilde{\xi}^{(\pi)}_{\pi(i)} = \xi^{(\pi)}_{\pi(i)}$ for $i \in \{1,\dots,R_{\{l_j\}}\}$ are kept free, where we have omitted the $\{l_j\}$ dependence of the permutation $\pi$ to simplify the notation. Let us remark that it holds:

$$a(\tilde{\xi}^{(\pi)}_{\pi(h)}|\{\tilde{\xi}^{(\pi)}_{j\leq N}\},\eta) = 0, \quad \forall h \in \{R_{\{l_j\}}+1,\dots,N-1\} \text{ and } a(\tilde{\xi}^{(\pi)}_{\pi(N)}|\{\tilde{\xi}^{(\pi)}_{j\leq N}\},\eta) \neq 0, \tag{B.53}$$

then defined:

$$\xi^{(\pi)}_{\pi(a)}(\epsilon) = \tilde{\xi}^{(\pi)}_{\pi(a)} + a\epsilon, \qquad \forall a \in \{1,\dots,N\}, \tag{B.54}$$

when $\epsilon$ approaches zero it holds:

$$-\frac{t_{l_{\pi(h)}}(\xi^{(\pi)}_{\pi(h)}(\epsilon)|\{\xi^{(\pi)}_{j\leq N}(\epsilon)\},\{k_i=\delta_{i,1}\},\eta)}{a(\xi^{(\pi)}_{\pi(h)}(\epsilon)|\{\xi^{(\pi)}_{j\leq N}(\epsilon)\},\eta)} = \epsilon^{-1}\hat{t}^{(\pi|\{l_i\})}_h(\xi_{N+1},\eta) +$$

$$+ \tilde{t}^{(\pi|\{l_i\})}_h(\xi_{N+1},\eta) + O(\epsilon), \tag{B.55}$$

where

$$(\hat{t}^{(\pi|\{l_i\})}_h(\xi_{N+1},\eta), \tilde{t}^{(\pi|\{l_i\})}_h(\xi_{N+1},\eta)) \neq (0,0) \quad h \in \{R_{\{l_j\}}+1,\dots,N\}, \tag{B.56}$$

$$\text{and} \qquad \hat{t}^{(\pi|\{l_i\})}_N(\xi_{N+1},\eta) = 0. \tag{B.57}$$

So defining:

$$t^{(\pi|\{l_i\})}_h(\xi_{N+1},\eta) = \begin{cases} \hat{t}^{(\pi|\{l_i\})}_h(\xi_{N+1},\eta) \text{ if this is nonzero} \\ \tilde{t}^{(\pi|\{l_i\})}_h(\xi_{N+1},\eta) \text{ otherwise} \end{cases}, \quad \forall h \in \{R_{\{l_j\}}+1,\dots,N\}, \tag{B.58}$$

it holds:

$$t^{(\pi|\{l_i\})}_h(\xi_{N+1},\eta) \neq 0, \quad \forall h \in \{R_{\{l_j\}}+1,\dots,N\}. \tag{B.59}$$

We can now observe that the following block structure emerges:

$$[c_{t_{l_1}(\xi^{(\pi)}_1(\epsilon)),\dots,t_{l_N}(\xi^{(\pi)}_N(\epsilon))}(\xi_{N+1})]_{\pi(a),\pi(b)} = x^{(\pi)}_{a,b}(\epsilon),$$
$$\forall (a,b) \in \{1,\dots,R_{\{l_j\}}\} \times \{1,\dots,R_{\{l_j\}}\}, \tag{B.60}$$

$$[c_{t_{l_1}(\xi^{(\pi)}_1(\epsilon)),\dots,t_{l_N}(\xi^{(\pi)}_N(\epsilon))}(\xi_{N+1})]_{\pi(a),\pi(b)} = \delta_{a+1,b}x^{(\pi)}_{a,a+1}(\epsilon) + \epsilon(1-\delta_{a+1,b})y^{(\pi)}_{a,b}(\epsilon),$$
$$\forall (a,b) \in \{1,\dots,R_{\{l_j\}}\} \times \{R_{\{l_j\}}+1,\dots,N\}, \tag{B.61}$$

$$[c_{t_{l_1}(\xi^{(\pi)}_1(\epsilon)),\dots,t_{l_N}(\xi^{(\pi)}_N(\epsilon))}(\xi_{N+1})]_{\pi(a),\pi(b)} = \epsilon y^{(\pi)}_{a,b}(\epsilon),$$
$$\forall (a,b) \in \{R_{\{l_j\}}+1,\dots,N\} \times \{1,\dots,R_{\{l_j\}}\}, \tag{B.62}$$

$$[c_{t_{l_1}(\xi^{(\pi)}_1(\epsilon)),\dots,t_{l_N}(\xi^{(\pi)}_N(\epsilon))}(\xi_{N+1})]_{\pi(a),\pi(b)} = \left(\epsilon^{-1}\hat{t}^{(\pi|\{l_i\})}_h(\xi_{N+1},\eta) + \tilde{t}^{(\pi|\{l_i\})}_h(\xi_{N+1},\eta) + O(\epsilon)\right) +$$
$$+ \delta_{a+1,b}x^{(\pi)}_{a,a+1}(\epsilon) + \epsilon(1-\delta_{a+1,b})y^{(\pi)}_{a,b}(\epsilon),$$
$$\forall (a,b) \in \{R_{\{l_j\}}+1,\dots,N\} \times \{R_{\{l_j\}}+1,\dots,N\}, \tag{B.63}$$

where it holds

$$x_{a,b}^{(\pi)}(0) = \prod_{\substack{c=1 \\ c \neq b}}^{N+1} \frac{\check{\xi}_{\pi(a)}^{(\pi)} - \check{\xi}_{\pi(c)}^{(\pi)} - \eta}{\check{\xi}_{\pi(b)}^{(\pi)} - \check{\xi}_{\pi(c)}^{(\pi)}}, \qquad y_{a,b}^{(\pi)}(0) = \prod_{\substack{c=1 \\ c \neq b,a+1}}^{N+1} \frac{\check{\xi}_{\pi(a)}^{(\pi)} - \check{\xi}_{\pi(c)}^{(\pi)} - \eta}{\check{\xi}_{\pi(b)}^{(\pi)} - \check{\xi}_{\pi(c)}^{(\pi)}}, \qquad (B.64)$$

and clearly

$$x_{a,a+1}^{(\pi)}(0) \neq 0 \text{ and } y_{a,b}^{(\pi)}(0) \neq 0 \quad \forall a, b \in \{1, \dots, N\}. \qquad (B.65)$$

Let us denote with $S_{\{l_i\}}$ the nonnegative integer defining the total number of $\hat{t}_h^{(\pi|\{l_i\})}(\xi_{N+1}, \eta) \neq 0$, then the next limit follows:

$$\lim_{\epsilon \to 0} \epsilon^{S_{\{l_i\}}} f_{t_{l_1}(\xi_1^{(\pi)}(\epsilon)),\dots,t_{l_N}(\xi_N^{(\pi)}(\epsilon))}(\xi_{N+1}, \eta)\Big|_{k_i = \delta_{1,i}} = \prod_{a=1}^{N} \prod_{\substack{b=1 \\ b \neq a}}^{N+1} \left( \check{\xi}_{\pi(a)}^{(\pi)} - \check{\xi}_{\pi(b)}^{(\pi)} \right)$$

$$\times \prod_{h=R_{\{l_j\}}+1}^{N} t_h^{(\pi|\{l_i\})}(\xi_{N+1}, \eta) \, \det_{R_{\{l_j\}}} \left[ [c_{t_{l_1}(\check{\xi}_1^{(\pi)}),\dots,t_{l_N}(\check{\xi}_N^{(\pi)})}(\xi_{N+1})]_{\pi(a),\pi(b)} \right] \neq 0, \quad (B.66)$$

being

$$\det_{R_{\{l_j\}}} \left[ [c_{t_{l_1}(\check{\xi}_1^{(\pi)}),\dots,t_{l_N}(\check{\xi}_N^{(\pi)})}(\xi_{N+1})]_{\pi(a),\pi(b)} \right] = \prod_{a=1}^{R_{\{l_j\}}} \frac{\prod_{b=1}^{N+1}(\check{\xi}_{\pi(a)}^{(\pi)} - \check{\xi}_{\pi(b)}^{(\pi)} - \eta)}{\prod_{\substack{b=1 \\ b \neq a}}^{N+1}(\check{\xi}_{\pi(a)}^{(\pi)} - \check{\xi}_{\pi(b)}^{(\pi)})}$$

$$\times \det_{R_{\{l_j\}}} \left[ \frac{1}{\check{\xi}_{\pi(a)}^{(\pi)} - \check{\xi}_{\pi(b)}^{(\pi)} - \eta} \right] \qquad (B.67)$$

$$= \prod_{a=1}^{R_{\{l_j\}}} \prod_{b=R_{\{l_j\}}+1}^{N+1} \frac{\check{\xi}_{\pi(a)}^{(\pi)} - \check{\xi}_{\pi(b)}^{(\pi)} - \eta}{\check{\xi}_{\pi(a)}^{(\pi)} - \check{\xi}_{\pi(b)}^{(\pi)}} \neq 0. \qquad (B.68)$$

This result implies that $f_{t_{l_1}(\xi_1^{(\pi)}(\epsilon)),\dots,t_{l_N}(\xi_N^{(\pi)}(\epsilon))}(\xi_{N+1}, \eta)\Big|_{k_i = \delta_{1,i}}$ is a nonzero Laurent polynomial of $\epsilon$, so that the polynomial (B.47) is not identically zero and so:

$$f_{t_{l_1}(\xi_1),\dots,t_{l_N}(\xi_N)}(\xi_{N+1}, \eta)\Big|_{k_i = \delta_{1,i}} \neq 0 \qquad (B.69)$$

holds almost for any choice of the parameters $\xi_{N+1}$, $\{\xi_{j \leq N}\}$ and $\eta$. Being this results true for any $\{l_1, \dots, l_N\} \in \{1, \dots, n^N\}^{\otimes N}$ the Lemma B.1 indeed implies that

$$f_{t_{l_1}(\xi_1),\dots,t_{l_N}(\xi_N)}(\xi_{N+1}, \eta) \neq 0 \qquad (B.70)$$

holds also for almost any choice of all the parameters $\xi_{N+1}$, $\{\xi_{j \leq N}\}$, $\{k_i\}$ and $\eta$ which completes our proof.

Let us compute now similarly the function

$$g_{t_{l_1}(\xi_1),\dots,t_{l_N}(\xi_N),j}(\xi_{N+1}, \eta)\Big|_{k_i = \delta_{1,i}}. \qquad (B.71)$$

In order to do so let us remark that defined:

$$\gamma(j) = N + 1, \quad \gamma(a) = a \qquad \forall a \in \{1, \dots, N\} \backslash \{j\}, \qquad (B.72)$$

we have the identity:

$$\det_N\left([c^{(j)}_{y_1,\ldots,y_j,\ldots,y_N}(\xi_{N+1})]_{ab}\right)=-\det_N\left([\bar{c}^{(j)}_{\bar{y}_1,\ldots,\bar{y}_j,\ldots,\bar{y}_N}(\xi_{N+1})]_{ab}\right),\tag{B.73}$$

where we have defined:

$$\bar{y}_h=y_h(1-\delta_{h,j}),\tag{B.74}$$

and

$$[\bar{c}^{(j)}_{y_1,\ldots,y_N}(\xi_{N+1})]_{ab}=-\delta_{ab}\frac{\bar{y}_a}{k_1a(\xi_a)}+\prod_{\substack{c=1\\c\neq\gamma(b)}}^{N+1}\frac{\xi_a-\xi_c-\eta}{\xi_{\gamma(b)}-\xi_c}\qquad\forall a,b\in\{1,\ldots,N\}.\tag{B.75}$$

So that we can compute $g_{t_{l_1}(\xi_1),\ldots,t_{l_N}(\xi_N),j}(\xi_{N+1},\eta)\big|_{k_i=\delta_{1,i}}$ exactly along the same lines we have computed $f_{t_{l_1}(\xi_1),\ldots,t_{l_N}(\xi_N)}(\xi_{N+1},\eta)\big|_{k_i=\delta_{1,i}}$ at least around some special point that we are going to define. From the equation (B.46) it follows that for any $\{l_1,\ldots,\widehat{l_j},\ldots,l_N\}\in\{1,\ldots,n^N\}^{\otimes(N-1)}$ there exist a $R_{\{l_j\}}\leq N$ and a $\pi_{\{l_j\}}\in S_N$, such that fixed $\pi_{\{l_j\}}(R_{\{l_j\}})=j$, it holds:

$$t_{l_{\pi_{\{l_j\}}(i)}}(\xi_{\pi_{\{l_j\}}(i)}|\{\xi_{j\leq N}\},\{k_i=\delta_{1,i}\},\eta)=0\quad\text{if }i\leq R_{\{l_j\}}-1\tag{B.76}$$

$$t_{l_{\pi_{\{l_j\}}(i)}}(\xi_{\pi_{\{l_j\}}(i)}|\{\xi_{j\leq N}\},\{k_i=\delta_{1,i}\},\eta)\neq0\quad\text{if }R_{\{l_j\}}+1\leq i.\tag{B.77}$$

Then we define:

$$\tilde{\xi}^{(\pi)}_{\pi(a+1)}=\xi_{N+1}-(N-a)\eta,\qquad\forall a\in\{R_{\{l_j\}},\ldots,N-1\},\tag{B.78}$$

while the $\tilde{\xi}^{(\pi)}_{\pi(i)}=\xi^{(\pi)}_{\pi(i)}$ for $i\in\{1,\ldots,R_{\{l_j\}}\}$ are kept free, where we have omitted the $\{l_j\}$ dependence of the permutation $\pi$ to simplify the notation. From this point we can proceed as for $f_{t_{l_1}(\xi_1),\ldots,t_{l_N}(\xi_N)}(\xi_{N+1},\eta)\big|_{k_i=\delta_{1,i}}$ and we get:

$$\lim_{\epsilon\to0}\epsilon^{S_{\{l_i\}}}g_{t_{l_1}(\xi^{(\pi)}_1(\epsilon)),\ldots,t_{l_N}(\xi^{(\pi)}_N(\epsilon)),j}(\xi_{N+1},\eta)=\prod_{a=1}^{N}\prod_{\substack{b=1\\b\neq a}}^{N+1}(\tilde{\xi}^{(\pi)}_a-\tilde{\xi}^{(\pi)}_b)\prod_{h=R_{\{l_j\}}+1}^{N}t^{(\pi|\{l_i\})}_h(\xi_{N+1},\eta)$$

$$\times\det_{R_{\{l_j\}}}\left[[\bar{c}^{(j)}_{t_{l_1}(\tilde{\xi}^{(\pi)}_1),\ldots,t_{l_N}(\tilde{\xi}^{(\pi)}_N)}(\xi_{N+1})]_{\pi(a),\pi(b)}\right]\neq0,\quad\text{(B.79)}$$

indeed:

$$\det_{R_{\{l_j\}}}\left[[\bar{c}^{(j)}_{t_{l_1}(\tilde{\xi}^{(\pi)}_1),\ldots,t_{l_N}(\tilde{\xi}^{(\pi)}_N)}(\xi_{N+1})]_{\pi(a),\pi(b)}\right]$$

$$=\prod_{a=1}^{R_{\{l_j\}}}\frac{\prod_{b=1}^{N+1}(\tilde{\xi}^{(\pi)}_{\pi(a)}-\tilde{\xi}^{(\pi)}_b-\eta)}{\prod_{\substack{b=1\\b\neq\gamma(\pi(a))}}^{N+1}(\tilde{\xi}^{(\pi)}_{\gamma(\pi(a))}-\tilde{\xi}^{(\pi)}_b)}\det_{R_{\{l_j\}}}\left[\frac{1}{\tilde{\xi}^{(\pi)}_{\pi(a)}-\tilde{\xi}^{(\pi)}_{\pi(b)}-\eta}\right]\tag{B.80}$$

$$=\frac{\prod_{b=1}^{R_{\{l_j\}}-1}(\xi_j-\tilde{\xi}^{(\pi)}_{\pi(b)})}{\prod_{b=1}^{N}(\xi_{N+1}-\tilde{\xi}^{(\pi)}_b)}\prod_{a=1}^{R_{\{l_j\}}-1}\prod_{b=R_{\{l_j\}}+1}^{N+1}\frac{\tilde{\xi}^{(\pi)}_{\pi(a)}-\tilde{\xi}^{(\pi)}_{\pi(b)}-\eta}{\tilde{\xi}^{(\pi)}_{\pi(a)}-\tilde{\xi}^{(\pi)}_{\pi(b)}}\neq0,\tag{B.81}$$

which completes the proof of our proposition. $\qquad\square$

It is interesting to remark that the rational functions $f_{t_l}(\xi_{N+1}, \{\xi_{j\leq N}^{(\pi)}\}, \{k_i = \delta_{1,i}\}, \eta)$ and $g_{t_l,j}(\xi_{N+1}, \{\xi_{j\leq N}^{(\pi)}\}, \{k_i = \delta_{1,i}\}, \eta)$ admits some simple explicit expression as a consequence of the calculations developed in the previous proposition.

*Corollary* B.2. *There exist a $R_l \leq N$ and a $\pi_l \in S_N$ such that:*

$$\lim_{k_i \to \delta_{1,i}} t_l(\lambda|\{k_i\}) = \prod_{a=1}^{R_l}(\lambda - \xi_{\pi_l(a)}) \prod_{a=1+R_l}^{N}(\lambda - \xi_{\pi_l(a)} + \eta), \tag{B.82}$$

*where $t_{l\leq n^N}(\lambda|\{k_i\})$ is a generic eigenvalue of the transfer matrix. Then, it holds:*

$$f_{t_l}(\xi_{N+1}, \{\tilde{\xi}_{j\leq N}^{(\pi_l)}\}, \{k_i = \delta_{1,i}\}, \eta) = \prod_{a=1}^{N}\prod_{\substack{b=1\\b\neq a}}^{N}\left(\tilde{\xi}_a^{(\pi_l)} - \tilde{\xi}_b^{(\pi_l)}\right) \prod_{a=R_l+1}^{N}\left(\tilde{\xi}_{\pi_l(a)}^{(\pi_l)} - \tilde{\xi}_{N+1}^{(\pi_l)}\right)$$

$$\times \prod_{a=1}^{R_l}\left(\tilde{\xi}_{\pi_l(a)}^{(\pi_l)} - \tilde{\xi}_{N+1}^{(\pi_l)} - \eta\right), \tag{B.83}$$

*for any $j \in \{1,\ldots,N\}$, where we have defined*

$$\tilde{\xi}_{\pi(a+1)}^{(\pi)} = \xi_{N+1} - (N-a)\eta, \qquad \forall a \in \{R_l, \ldots, N-1\}, \tag{B.84}$$

*while the $\tilde{\xi}_{\pi(i)}^{(\pi)}$ for $i \in \{1,\ldots,R_l\}$ are kept free.*

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
