# Peer review of "Complete spectrum of quantum integrable lattice models associated to Y(gl(n)) by separation of variables"

_SciPost Physics, doi:SciPost Phys. 6, 071 (2019)_

## Round 2 · Referee Report · Thiago Silva Tavares (Referee 2) · 2018-12-20

Strengths

Among the merits of the paper, the authors provide very technical proofs to a series of bijections which indeed completely characterize the spectrum of the model at hand. The proof for the $gl_n$ case is rather lengthier than the simpler $gl_2$ case and shows how one could compute the algebraic structure of the Bethe algebra.
It seems to me this is an important step to our understanding of integrability in quantum systems. Therefore I would like to recommend the manuscript for a publication in SciPost Physics.

Weaknesses

No weaknesses

Report

This manuscript is an extension of [1] where the authors developed a new scheme for the separation of variables for quantum integrable systems. Namely, acting with the transfer matrix calculated at the inhomogeneity parameters on a generic co-vector, one may generate a separate basis under the hypothesis that there exists one point of the integrable manifold (or a rearrangement of conserved charges) where the associated transfer matrix is nonderogatory. In other words, the transfer matrix is basis generating. As a consequence, one finds very simple wave-function in covector basis given either by product of powers of the transfer matrix eigenvalue at inhomogeneity parameters or powers of the Q-operator eigenvalue at inhomogeneity parameters and inhomogeneity parameters shifted by crossing parameter.
The manuscript extended [1] in that the authors apply the method to the fundamental representation for the $gl_n$ invariant solution of the Yang-Baxter equation. Important to the previous and to the present work is the possibility to recover the transfer matrix at any spectral parameter by means of an interpolation due to: the analytical dependence on the spectral parameter (polynomial for definiteness), asymptotic behavior, and fusion hierarchy specified at inhomogeneity parameters. Hence the complete spectrum is characterized in two different ways:
- by means of a system of $N$ algebraic equations of order $n$, $N$ being the system size or by spectral curve equation. (section 3).
- by means of a spectral curve, a difference functional equation of order $n$ (section 4).

Among the merits of the paper, the authors provide very technical proofs to a series of bijections which indeed completely characterize the spectrum of the model at hand. The proof for the $gl_n$ case is rather lengthier than the simpler $gl_2$ case and shows how one could compute the algebraic structure of the Bethe algebra.
It seems to me this is an important step to our understanding of integrability in quantum systems. Therefore I would like to recommend the manuscript for a publication in SciPost Physics. Before that, however, I suggest a few amendments:
- In page 5, first paragraph, and page 10, the paragraph after equation (3.13), the authors state that any operator commuting with a w-simple operator can be written as a polynomial of it, the maximum degree being the dimension d of the Hilbert space. Actually, as the author pointed out in [1], the maximum degree is d-1.

- In page 5, third paragraph, line 8: “There it was however pointed out that…” -> However, it was pointed out there that…

- I suggest to fully reserve the index n to the “rank” of the group $gl_n$. See, for instance, equation (3.11) and (4.1).

- If it is not much of work, I suggest dividing (4.18) by $\alpha_1(\xi_a)$. Also in (4.20) let only $\varphi$ on the right hand site.

- I believe that in equation (4.22) the function $a(\xi_a)$ has not been defined so far.

- In equation (5.3), page 15, I there is a misprint in the left hand side. The index should be $m+1$.

- In equation (A.16), page 19, there is a misprint in the index of the Kronecker delta.

- After equation (A.23), page 20, N-tupla -> N-tuple.

I do also have some questions which may contribute to further discussion if the authors wish include it, or to just respond it.
- In general, how one may find the spectral curve functional of equation (4.3) in a constructive way? Does SoV provide a means to that, similar to algebraic Bethe ansatz, or one should guess it by means, for instance, of off-diagonal Bethe ansatz.
- It seems to me that there exists a trade-off between the concepts of w-simplicity (in which one should have generic inhomogeneity parameters with w-simple twist matrix) and local conserved charges (when ideally we have homogeneous system). Integrability is not lost in the latter case. How can one harmonize these two situations? Does SoV provide a description of Bethe vectors in the homogeneous lattice?

[1] J. Math. Phys. 59 (2018) 091477

Requested changes

I suggest a few amendments:
- In page 5, first paragraph, and page 10, the paragraph after equation (3.13), the authors state that any operator commuting with a w-simple operator can be written as a polynomial of it, the maximum degree being the dimension d of the Hilbert space. Actually, as the author pointed out in [1], the maximum degree is d-1.

- In page 5, third paragraph, line 8: “There it was however pointed out that…” -> However, it was pointed out there that…

- I suggest to fully reserve the index n to the “rank” of the group $gl_n$. See, for instance, equation (3.11) and (4.1).

- If it is not much of work, I suggest dividing (4.18) by $\alpha_1(\xi_a)$. Also in (4.20) let only $\varphi$'s on the right hand site.

- I believe that in equation (4.22) the function $a(\xi_a)$ has not been defined so far.

- In equation (5.3), page 15, I there is a misprint in the left hand side. The index should be $m+1$.

- In equation (A.16), page 19, there is a misprint in the index of the Kronecker delta.

- After equation (A.23), page 20, N-tupla -> N-tuple.

I do also have some questions which may contribute to further discussion if the authors wish.
- In general, how one may find the spectral curve functional of equation (4.3) in a constructive way? Does SoV provide a means to that, similar to algebraic Bethe ansatz, or one should guess it by means, for instance, of off-diagonal Bethe ansatz.
- It seems to me that there exists a trade-off between the concepts of w-simplicity (in which one should have generic inhomogeneity parameters with w-simple twist matrix) and local conserved charges (when ideally we have homogeneous system). Integrability is not lost in the latter case. How can one harmonize these two situations? Does SoV provide a description of Bethe vectors in the homogeneous lattice?

---

## Round 2 · Referee Report · Anonymous (Referee 1) · 2018-12-20

Strengths

1- General treatment of $Y(gl_n)$ models with fundamental representations and general twisted boundary conditions 2- Good illustration of the new (original) approach to SoV method, developed by the authors

Weaknesses

1- Proofs are rather technical, and the redaction could be improved

Report

The authors apply the "improved" SoV method they have developed recently to the case of integrable spin chains based on the Yangian $Y(gl_n)$.
More precisely, they consider a N-sites lattice with fundamental representations and inhomogeneities on each site, and twisted periodic boundary conditions. The twist matrix is generic (but not zero) in the sense the calculation uses only its Jordanian decomposition, with no assumption on its diagonalizability.
Their previous article [1] (arXiv:1807.11572) was presenting the method, using mainly the $gl_2$ case to illustrate it. The general case of the Yangian $Y(gl_n)$ is done in the present paper.

The basic idea (introduced in [1]), is to use the Bethe algebra (the set of fused transfer matrices) to construct a covector basis adapted to the SoV method (developed by Sklyanin in the 90's). The method is rather universal and allows a general treatment which is less model dependent (which was lacking in the original approach by Sklyanin, although Sklyanin gave algebraic characterizations for large classes of models). It uses fusion relations of the transfer matrices and interpolation between specific points (where the spectral parameter is equal to one of the inhomogeneity, or asymptotic behavior) to fully characterize the spectrum of the transfer matrices and prove its simplicity. The spectrum is characterized by a quantum spectral curve, obtained as a difference functional equation of order $n$. The decomposition of the eigenvectors in the SoV basis is also computed, the coefficients being given in term of the eigenvalues. When the twist matrix is diagonalizable, with simple spectrum, they also construct the Baxter's Q-operator and show that it obeys T-Q relations of order $n$.

The paper is interesting and sounded. It clearly deserves to be published.
However, I think that there are few points that could be improved. In particular, since the proofs presented in the manuscript are rather technical, the authors should try to ease the reading. I give a list of suggestions below. Once it is done, the paper can be published.

Requested changes

1- At the end of the second paragraph of the introduction (page 3), one should not say that the algebraic Bethe ansatz fails. It has to be improved (in the same way the author have improved the SoV method), and the general framework has being given in a series of articles for the XXZ models (arXiv:1506.02147, arXiv:1412.7511, arXiv:1408.4840). It is misleading to say that it fails. The twisted XXX model has being also studied in this way in arXiv:1805.11323 and arXiv:1804.00597, and a fair treatment requires to cite them.

2- In the first sentence of the third paragraph (this sentence being rather long, in passing), it is not clear whether $Y_n$ denotes the eigenvalues or the eigenvectors. It is a bit clarified by the second sentence (of the same length) but I think that the last $Y_n$ in this sentence should be $B(\lambda)$. Shorter sentences would be better for the understanding.

3- Second paragraph of page 5, speaking of analytical Bethe ansatz and fusion, they should cite the paper arXiv:math-ph/0411021.

4- Beginning of page 7, they should add that they shift the notation from $T^{(K)}(\lambda,{\xi})$ to $T^{(K)}(\lambda)$. Since there is a lot of notation around these transfer matrices $T$, I think it will help a lot to clearly states the notation.

5- Proposition 3.1 (page 9): the notion of w-simple matrix is not defined. It is true that it is given in [1], but it should be reminded here. Still in this proposition, the use of the same index $a$ to label either the sites ($a=1...N$) or the Jordan block of $K(\lambda)$ ($a=1...M$) does not help: another letter (like $j$, which is used later on to label the eigenvalues $k_j$ of $K(\lambda)$) would be better. In the same way, calling $x$ the entries of the covector $<S|$ (3.5) while the same letter $x$ is used right after to denote a generic point or a point in $\Sigma_T$ (theorem 3.1) is not the best choice. Since the covector $<S|$ essentially relies on $W_{K,a}$, I would suggest the letter $w$ for instance.

6- Corollary 4.1 (page 14): they have to mention that $k_j$ are the eigenvalues of $K(\lambda)$. In eq (4.22) there is a function $a$ which is not defined. Using $a$ to denote polynomials and at the same time using $a$ as an index is not a good move. There are plenty of letters in the alphabet, play with them. The same remark also applies to the appendix B (see below).

7- A short conclusion is missing. For instance, the authors could comment on the limit $\xi_a\to 0$: since the interpolation relies on the fact that all $\xi$'s are different, what can we say about the homogeneous limit? Is the thermodynamical limit easy to deal with in this framework?

8- In the proof of lemma A.1, I think it would be clearer if they first do the full proof for $N^+_m=0$ before starting the proof for $N^-_m=0$. This amounts to move eq. (A.7) after eq (A.12) (when it is needed) and would make the proof more linear.

9- In lemma A.3 and the corresponding proof, I think that there is a mixing between the indices $b$ and $a$. For instance, in eq (A.63) $b$ is a fixed index (the one mentioned in the lemma), while at the beginning of page 24 it is a dummy index. In eq. (A.66) it seems that the fixed index is now $a$, while $b$ is the running one. All this makes the proof rather difficult to follow.

10- Middle of page 26, instead of $x={(n-1)/2 \text{ for $n$ odd}, (n-2)/2 \text{ for $n$ even}}$, why don't you just write $x=[n/2]-1$? To use $x$ again for an index at that stage is also a bit strange, but anyway, I guess it is a bit risky to change it everywhere.

11- Lemma B.2 (page 30): you have to write 'of order \textit{at most} M+R'. This is particularly true for the sum, whose order is closer to max(M,R) than M+R. In the proof of the lemma, eqs (B.14) and (B.15), indices $P$ and $Q$ are missing on the $C$'s.

12- Page 31, eq. (B.20), specify where $i$ runs. Up to now $n$ was corresponding to $Y(gl_n)$, and now it is a dummy index: change it as in eq. (B.21) where it is only partially done (a $n$ is still prowling around). What is $a(\xi)$? The same applies to eq (B.24).

13- Page 32, eqs B.30 and B.31: the function $c$ are not defined. Are they the same as the ones in (B.36) and (B.37), which have different indices?

14- Page 34, eq. (B.52), the index $a$ should run up to $N-1$ instead of $N$. Right after the equation there is a $R_l$ that should be $R_{{l_j}}$. These two remarks also apply to eq. (B.77).

Few typos (I guess the list is non-exhaustive):

15- After eq. (A.18), the second $\bar N^-_m$ should be $\bar N^+_m$.

16- After eq (A.43), I think $h_a$ should be $h_b$. Same thing after eq (A.48).

17- Page 27, 2nd line from the bottom: These $\to$ This

18- Page 28 last line of section A: form $\to$ from

19- Page 29, before eq (B.9): the sentence which starts with 'Indeed, denoted...' looks strange. At least, I think that 'denoted' should be 'denote', and it would be better to do shorter sentences.

20- Page 34, eq. (B.81), there a $R_h$ that should be $R_l$.

---

## Round 2 · Referee Report · Anonymous (Referee 3) · 2018-12-21

Strengths

  1. A radically new and very promising approach to separation of variables is proposed
  2. The new approach allows to circumvent the difficulties of the previous approach by Sklyanin, for gl(n) case.
  3. The technical difficulties of upscaling from gl(2) and gl(3) examples to generic gl(n) case are successfully overcome and the complete solution of the problem is presented.

Weaknesses

  1. Minor LaTeX formatting remarks, see in Requested changes
  2. Minor English errors, see in Requested changes

Report

In the previous paper, JMP 59, 091417 (2018), the authors have outlined a promising new approach to the separation of variables (SoV) in quantum integrable models. The idea of generalised SoV for the models whose integrability stems from an R-matrix algebra has been proposed in 1992-1996 by Sklyanin, however, his approach, though quite successful for gl(2)-type models, meets serious difficulties in application to gl(n)-type models. The new approach by Maillet and Niccoli is based on the direct construction of SoV basis by the repated action of transfer matrices on a fixed vector in Hilbert space, and is able to overcome the difficulties inherent in Sklyanin's approach.
In their first paper, the authors outlined the main ideas of the method and illustrated it on gl(2) and gl(3) case. In the present paper, 2nd in the row, the same approach is transferred to the generic gl(n) case. The upscaling is by no means automatic and requires a serious work with yangian quantum group. The authors have shown a lot of ingenuity and managed to successfully overcome the numerous technical difficulties to achieve a complete characterisation of the spectrum of the commuting transfer matrices in terms of the quantum spectral curve. A long-standing problem in quantum integrability is thus satisfactorily solved. I have no doubt that the new approach to SoV will lead to further breakthroughs and will look forward for the new developments.

Requested changes

  1. LaTex remark: Mathematical operations like End, tr, etc should be typed in upright \rm or \mathrm font, using \mathop command. See End in formula (2.1), tr in (2.6), q-det in (2.11) and in many other places.

  2. English: 1st line on p8: The general fusion identity [12,13,75] reduce --- choose either singular, or plural: identity/reduces, or identities/reduce. I think I saw a similar mistake a couple of times more but lost where.

---

## Round 3 · Referee Report · Anonymous (Referee 1) · 2019-5-30

Report

The authors have answered to the questions I raised in a positive way
Their manuscript can be published

---

## Round 3 · Referee Report · Thiago Silva Tavares (Referee 2) · 2019-5-31

Strengths

Among the merits of the paper, the authors provide very technical proofs to a series of bijections which indeed completely characterize the spectrum of the model at hand. The proof for the $gl_n$ case is rather lengthier than the simpler $gl_2$ case and shows how one could compute the algebraic structure of the Bethe algebra. It seems to me this is an important step to our understanding of integrability in quantum systems. Therefore I would like to recommend the manuscript for a publication in SciPost Physics.

Weaknesses

No Weaknesses

Report

I would like to thank the authors for their response. They have addressed the raised points in a very satisfactory manner. Nevertheless, I would like include some other minor modifications to improve the writing of the manuscript. The list is suggestive.

Requested changes

Introduction, page 3, paragraph 1, line 2:
variables that we recently developed -> variables that we developed recently

Introduction, page 3, paragraph 2, line 2:
by the use -> by using

Introduction, page 4, paragraph 1, line 8:
quantum model -> quantum models

Introduction, page 4, paragraph 3, line 1:
allows us -> allows

Introduction, page 4, paragraph 3, line 5:
made coinciding -> made to coincide

Introduction, page 4, paragraph 3, line 6:
applies for -> applies to

Introduction, page 4, paragraph 4, line 18:
satisfying with the transfer matrices the same $n$ order… -> satisfying along with the transfer matrices the same order $n$…

Introduction, page 5, paragraph 1, line 19:
characterization -> characterizations

Introduction, page 5, paragraph 3, line 8:
it was there pointed out that -> it was pointed out there that

Section 2, page 6, just before (2.2):
it is solution -> it is a solution

Section 2, page 6, just before (2.5):
I prefer to call (2.5) the fundamental relation or just the Yang-Baxter algebra instead of calling it Yang-Baxter equation.

Section 2, page 7, just before (2.7):
… they are in generic position -> … they are in generic positions

Section 3, page 9, equation (3.9):
Maybe it would be better to change the name of the set $\{x_1,\ldots,x_N\}$. Maybe just $\Sigma$ without $T$.

Section 4, page 11, theorem 4.1:
Then an entire functions -> Then an entire function

Section 4, page 12, just before equation (4.9):
satisfies -> satisfy

Section 4, page 13, equation (4.20):
In equation (4.20) I suggest to let only $\varphi$'s on the RHS, in such a way that either you obtain the separate basis in terms of the transfer matrix with Lax operators normalized to be unitary (LHS), or by a specific combination of $Q$ operators (RHS).

Section 4, page 14, Corollary 4.1:
… let us denote with $k_j$ the corresponding eigenvalues -> let us denote $k_j$ the corresponding eigenvalues

Conclusion, page 18, paragraph 1, line 2:
Resolution -> solution

Conclusion, page 18, paragraph 2, line 13:
is true in this homogeneous limit -> is true in the homogeneous limit.

Conclusion, page 18, paragraph 2, line 14:
associated the -> associated to the

Conclusion, page 18, paragraph 2, line 21:
Is the fundamental first step -> is the first fundamental step

Appendix A, page 19, equation (A.12):
In this series of equations there is an index a with a different format

Appendix A, page 20, just before equation (A.17):
… in the case both -> in the case where both

Appendix A, page 21, just before equations (A.24)
N-tupla -> N-tuple

Appendix A, page 22, equations (A.36) and (A.41)
I believe that there is a misprint in the index of $N^+$. Instead of $m-1$ we should have $m+1$.

Appendix A, page 22, just after equation (A.48)
I believe that there is a misprint in the indices defining the loop

Appendix A, page 22, footnote, line 2:
Smaller of one unit of the one in the first step covectors -> one unit smaller than in the first step convectors

---

## Round 3 · Author Response

Dear Editor,

We are sincerely grateful to the referees for their careful reading of the manuscript, the interest shown in our work, their valuable remarks and moreover for the useful detection of several misprints.

We have implemented most of the referee’s requests. In the following, we give in detail the main modifications that we have done (besides, we also fixed some other minor misprints), and we answered the questions or remarks of the referees.

Both the referee’s 1 and 2 have asked about the homogeneous (and thermodynamic) limits. We have added a conclusion section to our paper, in particular to discuss this issue and also to give further comments on future developments. Here, let us just say that the general idea is to develop our analysis for both the spectrum and the dynamics in the inhomogeneous models and then to consider their homogeneous limit. This strategy has proven to be the most efficient in our previous works, in particular in the algebraic Bethe ansatz approach to correlation functions of quantum integrable lattice models, and we think this is also the best one in the present context. Indeed, considering first the inhomogeneous models enables one in general to give a simpler algebraic description of all the intermediate steps that would otherwise involve various (high) derivatives of the elementary objects involved. Moreover, let us remark that the characterization of the spectrum by TQ finite difference functional equation admits in general a smooth homogeneous limit. So, starting from the inhomogeneous complete characterization of the spectrum one gets the homogeneous one. However, one still has to prove that in the homogeneous limit this characterization is complete, i.e. that the full spectrum is described by the TQ equation. One way to prove it can be by direct construction of the SoV basis in the homogeneous models. This requires the proof of the non-degeneracy of the spectrum of the full set of fused transfer matrices in this limit. The other main subject which it is under analysis is the computation of scalar products and matrix elements on separate states for higher rank case. In the SoV framework for integrable quantum models related to gl(2), i.e. to rank one case, this type of analysis has been already developed with our collaborators N. Kitanine and V. Terras. Hence, the current analysis will represent a natural but non trivial extension of it.

The second referee asked also about a constructive derivation of the TQ equation in the SoV framework. We have added some comments about this in the Remark 4.1 after the Theorem 4.1 where this TQ equation is given. Let us say that from our point of view the way the TQ equation has been derived in our SoV framework is quite constructive once one follows some intuitions inherited from the classical case. The fused transfer matrices are by definition the “shifted” quantum analogue of the classical spectral invariants. So, it is natural to look for them (and for their eigenvalues) to satisfy a “shifted” quantum analogous of the spectral curve equation. Then, we are left just with the determination of the shifts in the argument of these transfer matrices and the computation of the corresponding coefficients in the TQ equations. These are indeed completely fixed by the known fusion relations and asymptotic behavior of the transfer matrices together with their known central zeroes. The second referee also referred to Algebraic Bethe Ansatz (ABA) to have a constructive derivation of the TQ equation. Indeed, this is quite natural in the rank one case along the lines we just described. Moreover, Sklyanin has shown that the Nested ABA form of the eigenvalues of the fused transfer matrices of the rank 2 case can be also used to reconstruct the TQ equations. However, one should mention that ABA by itself gives only an ansatz for the form of the transfer matrix eigenvalues and eigenvectors. The main problem in the ABA framework is to prove that indeed all the eigenvalues have the given form in terms of Q-functions, i.e. to address the completeness problem which is solved in the SoV framework.

Concerning the first Referee’s requests we have implemented them in the following way. For the request 1, we have modified the sentences containing the reference to ABA and we have added a footnote with the papers cited by the referee. For the request 2, we have changed some sentences referring to $Y_n$ to make more clear that they are the operators zeros of the commuting family of operators $B(\lambda)$. All the other requests have been implemented with the exception of 10 and 16.
Concerning 10 our x coincides with [n/2] for n odd and [n/2]-1 for n even. Concerning 16, the correct one is an $h_a$. In fact, this $a$ refers to the index appearing in the equations (A.27)-(A.30).
Note that as pointed out between (A.26) and (A.27), $h_a$ in the original covector appearing in (A.25)-(A.26) is assumed to be an integer between 0 and n-m-1. So that in the first step covector, appearing in (A.27)-(A.28), $h_a$ is incremented by one unit and so, it is an integer between 1 and n-m. Now by definition the value of $N^{+}_{m}$ of the second step covectors, appearing in (A.42)-(A.43), is one unit smaller compared to the one of the first step covectors, which reads $\bar{N}_{m}^{+}+\delta _{n-m}^{h_{a}}$. This last value is computed using the definition of $N^{+}_{m}$, equation (A.30) and recalling that in the first step covectors the value of $h_{a}$ is one unit bigger compared to the value that it has in the original covectors. We have added a note to clarify this point after equation (A.43). Finally, we have defined the function $a(\lambda)$ in equation (4.5) as this was indeed missing.

Concerning the other requests from the second and the third referees we have implemented them all.

Sincerely yours,

J. M. Maillet
G. Niccoli

---

## Round 3 · List of Changes

Both the referee’s 1 and 2 have asked about the homogeneous (and thermodynamic) limits. We have added a conclusion section to our paper, in particular to discuss this issue and also to give further comments on future developments. Here, let us just say that the general idea is to develop our analysis for both the spectrum and the dynamics in the inhomogeneous models and then to consider their homogeneous limit. This strategy has proven to be the most efficient in our previous works, in particular in the algebraic Bethe ansatz approach to correlation functions of quantum integrable lattice models, and we think this is also the best one in the present context. Indeed, considering first the inhomogeneous models enables one in general to give a simpler algebraic description of all the intermediate steps that would otherwise involve various (high) derivatives of the elementary objects involved. Moreover, let us remark that the characterization of the spectrum by TQ finite difference functional equation admits in general a smooth homogeneous limit. So, starting from the inhomogeneous complete characterization of the spectrum one gets the homogeneous one. However, one still has to prove that in the homogeneous limit this characterization is complete, i.e. that the full spectrum is described by the TQ equation. One way to prove it can be by direct construction of the SoV basis in the homogeneous models. This requires the proof of the non-degeneracy of the spectrum of the full set of fused transfer matrices in this limit. The other main subject which it is under analysis is the computation of scalar products and matrix elements on separate states for higher rank case. In the SoV framework for integrable quantum models related to gl(2), i.e. to rank one case, this type of analysis has been already developed with our collaborators N. Kitanine and V. Terras. Hence, the current analysis will represent a natural but non trivial extension of it.

The second referee asked also about a constructive derivation of the TQ equation in the SoV framework. We have added some comments about this in the Remark 4.1 after the Theorem 4.1 where this TQ equation is given. Let us say that from our point of view the way the TQ equation has been derived in our SoV framework is quite constructive once one follows some intuitions inherited from the classical case. The fused transfer matrices are by definition the “shifted” quantum analogue of the classical spectral invariants. So, it is natural to look for them (and for their eigenvalues) to satisfy a “shifted” quantum analogous of the spectral curve equation. Then, we are left just with the determination of the shifts in the argument of these transfer matrices and the computation of the corresponding coefficients in the TQ equations. These are indeed completely fixed by the known fusion relations and asymptotic behavior of the transfer matrices together with their known central zeroes. The second referee also referred to Algebraic Bethe Ansatz (ABA) to have a constructive derivation of the TQ equation. Indeed, this is quite natural in the rank one case along the lines we just described. Moreover, Sklyanin has shown that the Nested ABA form of the eigenvalues of the fused transfer matrices of the rank 2 case can be also used to reconstruct the TQ equations. However, one should mention that ABA by itself gives only an ansatz for the form of the transfer matrix eigenvalues and eigenvectors. The main problem in the ABA framework is to prove that indeed all the eigenvalues have the given form in terms of Q-functions, i.e. to address the completeness problem which is solved in the SoV framework.

Concerning the first Referee’s requests we have implemented them in the following way. For the request 1, we have modified the sentences containing the reference to ABA and we have added a footnote with the papers cited by the referee. For the request 2, we have changed some sentences referring to $Y_n$ to make more clear that they are the operators zeros of the commuting family of operators $B(\lambda)$. All the other requests have been implemented with the exception of 10 and 16.
Concerning 10 our x coincides with [n/2] for n odd and [n/2]-1 for n even. Concerning 16, the correct one is an $h_a$. In fact, this $a$ refers to the index appearing in the equations (A.27)-(A.30).
Note that as pointed out between (A.26) and (A.27), $h_a$ in the original covector appearing in (A.25)-(A.26) is assumed to be an integer between 0 and n-m-1. So that in the first step covector, appearing in (A.27)-(A.28), $h_a$ is incremented by one unit and so, it is an integer between 1 and n-m. Now by definition the value of $N^{+}_{m}$ of the second step covectors, appearing in (A.42)-(A.43), is one unit smaller compared to the one of the first step covectors, which reads $\bar{N}_{m}^{+}+\delta _{n-m}^{h_{a}}$. This last value is computed using the definition of $N^{+}_{m}$, equation (A.30) and recalling that in the first step covectors the value of $h_{a}$ is one unit bigger compared to the value that it has in the original covectors. We have added a note to clarify this point after equation (A.43). Finally, we have defined the function $a(\lambda)$ in equation (4.5) as this was indeed missing.

Concerning the other requests from the second and the third referees we have implemented them all.

---

## Editorial Decision

published